# Mol-LLaMA: Towards General Understanding of Molecules in Large Molecular Language Model

**Dongki Kim**[1], **Wonbin Lee**[1], **Sung Ju Hwang**[1,2]

KAIST[1], DeepAuto.ai[2]

{cleverki, smilelwb01, sungju.hwang}@kaist.ac.kr

## Abstract

Understanding molecules is key to understanding organisms and driving advances in drug discovery, requiring interdisciplinary knowledge across chemistry and biology. Although large molecular language models have achieved notable success in task transfer, they often struggle to accurately analyze molecular features due to limited knowledge and reasoning capabilities. To address this issue, we present Mol-LLaMA, a large molecular language model that grasps the general knowledge centered on molecules and exhibits explainability and reasoning ability. To this end, we design key data types that encompass the fundamental molecular features, taking into account the essential abilities for molecular reasoning. Further, to improve molecular understanding, we propose a module that integrates complementary information from different molecular encoders, leveraging the distinct advantages of molecular representations. Our experimental results demonstrate that Mol-LLaMA is capable of comprehending the general features of molecules and providing informative responses, implying its potential as a general-purpose assistant for molecular analysis. Our project page is at https://mol-llama.github.io/.

## 1 Introduction

Understanding molecules and their properties is fundamental to understanding chemical compounds and living organisms, driving scientific discovery. However, it remains challenging due to the complexity of molecules and their behaviors, necessitating a comprehensive understanding of molecules. Recent advancements in large language models (LLMs) [46, 16, 39, 40] have demonstrated their potential in understanding core concepts in chemistry and biology [1, 43]. Further, LLMs have shown that language can play various roles, which is especially crucial to dealing with substantial complexity in chemistry, biology, and pharmacology by readily obtaining external knowledge from users' prompts. Nevertheless, LLMs struggle to interpret the molecular structures from the raw string representations such as SMILES [52] due to their nature of learning the subword representations.

To integrate structural information into LLMs, recent works have studied molecular LLMs that are instruction-tuned with the molecular modality, adopting the multi-modal instruction tuning widely studied in other modalities, such as image [24, 26], audio [13], and video [5, 60, 36]. In the multi-modal instruction tuning, there are two core components: 1) the molecular encoder to embed the molecular structures and 2) the instruction dataset to learn the molecule-related knowledge.

Despite their success, molecular LLMs still encounter difficulties in understanding and reasoning about the fundamental characteristics of molecules. Based on the observations from Table 1, we identify two key challenges to develop a general-purpose assistant for molecular analysis as follows:

- Molecular LLMs should learn the general features of molecules encompassing their chemical and biological features with a primary emphasis on the basic understanding of structures. Due to their limited knowledge, existing molecular LLMs often misinterpret molecular structures, incorrectly predicting key molecular features, or fail to provide meaningful explanations of molecular features.

39th Conference on Neural Information Processing Systems (NeurIPS 2025).

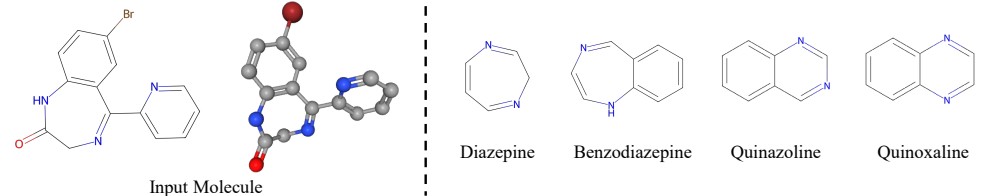

| | | | Diazepine | Benzodiazepine | Quinazoline | Quinoxaline |

Input Molecule

**Original Description**: Bromazepam binds to the GABA receptor GABA-A, causing a conformational change and *increasing inhibitory effects of GABA*. Other neurotransmitters are not influenced. One of the *benzodiazepines* that is *used in the treatment of anxiety disorders.*

**User**: Explain the features of the given molecule that are most distinct from those of other molecules.

**Mol-LLaMA (Ours)**

The given molecule is a member of the benzodiazepine class, which is characterized by a fusion of a benzene ring and a seven-membered diazepine ring. This structure is crucial for its biological activity, particularly its role as a sedative and hypnotic agent. ...<omitted>...

The given molecule is also characterized by a ketone group within the diazepine ring, which is a common feature in benzodiazepines. This functional group is important for the interaction with the GABA-A receptor, enhancing the inhibitory effects of GABA. ...<omitted>...

**GPT-4o**

1. **Quinazoline Core**: The molecule contains a quinazoline core, ...<omitted>...

Overall, the combination of a quinazoline structure with unique substitution patterns, including bromine halogenation and the presence of a pyridinyl group, ...<omitted>... .

**LLaMo**

The molecule has a benzodiazepine structure with a bromo substituent at the 7-position and a methyl group at the 1-position. It is distinct from other molecules in that it contains a bromine atom, a nitrogen atom, and a methyl group, which are not present in the other molecules.

**3D-MoLM**

...<omitted>... It belongs to the class of compounds called quinazolines, which are characterized by a bicyclic structure consisting of a benzene ring fused to a pyrimidine ring. ...<omitted>...

The molecule's structure suggests potential applications in medicinal chemistry, as quinazolines have been found to possess various biological activities, including antitumor, antimicrobial, and anti-inflammatory properties.

**Mol-Instructions**

The molecule is a quinoxaline derivative.

Table 1: Case study to compare molecular understanding and reasoning ability. Mol-LLaMA accurately understands the molecular features, answering a correct molecular taxonomy and providing its distinct properties that are relevant to the given molecule with detailed explanations. Entire responses are provided in Table 11.

- Explainability and reasoning capabilities are essential for molecular reasoning. Unlike other modalities where features are directly observable, such as image or video, molecular features are highly complex and are not readily apparent from structures. This complexity necessitates in-depth reasoning based on extensive domain knowledge and interpretability to serve as a general-purpose assistant. However, existing molecular LLMs often fail to provide rationales or perform a clear reasoning process, which undermines their ability to accurately predict molecular properties and makes it difficult to assess the reliability of their responses.

The shortcomings of recent molecular LLMs come from their limitations within each component of multi-modal instruction tuning. First, while the instruction tuning with relevant instruction datasets has been shown to expand knowledge and introduce reasoning capabilities [58], recent works [30, 3, 12, 61, 57] have primarily relied on public databases which contain knowledge and context restricted to specific tasks. Thus, the resulting molecular LLMs often generate irrelevant responses, making wrong predictions of molecular properties and exhibiting limited interpretability. Although the design of instruction datasets that entail domain-relevant knowledge and reasoning capabilities is crucial for molecular reasoning, it still remains underexplored. Second, existing molecular LLMs [30, 3, 41] rely on a single type of molecular encoder, limiting them to processing only one form of molecular structures, such as 2D or 3D molecular graphs. This shortcoming hinders

the accurate understanding and prediction of complex molecular features by misinterpretations of molecular structures.

To overcome the aforementioned limitations, we present Mol-LLaMA, a molecular LLM that learns fundamental knowledge centered on molecules with explainability and reasoning ability, positioning it as a general-purpose molecular assistant. To this end, we mainly explore the core components lying in multi-modal instruction tuning. First, we construct a novel instruction dataset by designing three key data types that encompass the fundamental features of molecules as well as cultivate the explainability and reasoning ability. Further, to improve the structural understanding, we propose a blending module that integrates complementary information from different molecular representations, alleviating the hallucination problem and enhancing the understanding of molecular features.

We experimentally validate the effectiveness of Mol-LLaMA in understanding the fundamental features of molecules for solving scientific problems, where Mol-LLaMA outperforms baselines including LLMs and molecular LLMs, providing accurate, detailed, and helpful responses. We further evaluate Mol-LLaMA on the molecular property prediction task and the molecular comprehension benchmarks, where it not only accurately predicts molecular properties but also generates relevant and helpful responses, highlighting its utility as a general-purpose assistant for molecular analysis. Our contributions can be summarized as follows:

- We present Mol-LLaMA that learns general knowledge for molecules across structural, chemical, and biological aspects and exhibits reasoning abilities and explainability, qualified as a general-purpose assistant for molecular analysis.

- We design three novel data types for the general understanding of molecules, establishing a large and informative instruction dataset centered on molecular features that can cultivate explainability and reasoning abilities of molecular LLMs.

- We devise a blending module to fully leverage the complementary information from different types of encoders, alleviating the hallucination and thus enhancing the understanding of molecular structures and advanced features.

- The proposed Mol-LLaMA outperforms previous LLMs and molecular LLMs including GPT-4o in the molecular understanding by learning the comprehensive knowledge centered on molecules.

## 2 Related Works

**Molecular Foundation Models**    Molecular foundation models have achieved remarkable success in modeling molecules using string representations [6, 11, 50, 20], 2D molecular graphs [56, 37], 3D molecular graphs [63, 32, 15, 53] or texts from biomedical literature [17, 23, 2]. Recently, with the emergence of molecule-text pair datasets, multi-modal foundation models have been developed based on contrastive learning [45, 29] or text decoders [10, 59, 34, 7, 28, 42]. However, these models remain limited to perform diverse tasks, which restricts to serve as general-purpose molecular assistants.

**Large Language Models on Scientific Discovery**    Large language models (LLMs) [39, 40, 46, 16, 4, 48, 8] have shown that they can play various roles via textual interactions with users. Recently, AI4Science and Quantum [1] have demonstrated the promising potential of LLMs in understanding a wide range of knowledge and solving complicated scientific problems. Despite their notable progress, LLMs struggle to interpret raw string representations such as SMILES [52] and SELFIES [22], as they learn subword representations that are not well-suited for tokenizing such raw string representations.

**Large Molecular Language Models**    Along with the remarkable progress in multi-modal large language models (LLMs) [27, 26, 55], molecular LLMs including MolCA [30], Mol-Instructions [12], LlasMol [57], InstructMol [3], and 3D-MoLM [25] have been developed by training LLMs on molecule-text pair datasets with graph representations modeled by 2D or 3D molecular encoder. In the concurrent work to ours, Park et al. [41] propose LLaMo with a projector to seamlessly encode the molecular structures from 2D representations, while training on the public databases and the constructed conversations. On the other hand, Luo et al. [35] propose BioMedGPT-LM to integrate the molecular and protein modalities into one LLM by training it on the public database. Despite their promising performance on task transfer, they are trained on the instruction datasets that are typically task-specific, hindering them from functioning as general-purpose assistants. In this work, we aim to build a molecular LLM capable of understanding of general features of molecules with explainability and reasoning capabilities to be utilized as a general-purpose assistant for molecular analysis.

# 3   Mol-LLaMA

We present Mol-LLaMA, a general-purpose assistant for molecular analysis that can comprehend the fundamental features of molecules with explainability and reasoning capabilities.

## 3.1   Instruction Dataset Construction for Comprehending Molecular Features

**Goal**   The instruction dataset lies at the core of the multi-modal instruction tuning, promoting understanding of other modalities and introducing the reasoning capabilities [27, 26, 58]. In the context of our work, scientific problems are interdisciplinary, necessitating wide-ranging expertise including chemistry and biology, along with complex reasoning skills. To this end, an ideal instruction dataset for molecular LLMs should encompass comprehensive information with detailed explanations of the underlying scientific principles.

However, existing instruction datasets lack comprehensive information. For example, in the case of the existing data example 1 of Table 2, the context of the target answer is highly limited, failing to provide relevant knowledge and an in-depth explanation of the causality of the target features. Moreover, the knowledge addressed in this data is restricted to specific tasks, hindering the learning of fundamental molecular features. Consequently, due to their narrow knowledge scope and limited reasoning capabilities, such instruction data obstructs molecular LLMs from functioning as general-purpose assistants. To overcome these limitations, our goal is to establish an instruction dataset that explicitly presents the fundamental features of molecules and explains their causality.

**Data Generation Pipeline for Molecular Reasoning**   To this end, we employ GPT-4o [40] whose capabilities for scientific knowledge and reasoning have been widely studied [1]. We prompt GPT-4o to generate instruction data by leveraging two types of contexts: 1) processed string representations of molecules and 2) their descriptions. For the processed string representations, we use the annotated IUPAC names, which explicitly specify the names of functional groups and their connectivity [14]. For the descriptions, we use annotated descriptions from PubChem [21] to provide grounded features.

The next question is how to design an instruction dataset that covers the fundamental features of molecules while cultivating the reasoning capabilities. To this end, we are inspired by the natural law, where the molecular features have a hierarchical relationship. That is, the structures determine the chemical features, while the biological features are determined by both the structural and chemical features. Based on our observation, we devise three novel data types that address these fundamental features as well as enhance the reasoning capabilities, as shown in Table 2.

1. **Detailed Structural Description:** We first design detailed descriptions of molecular structures to upskill the foundational understanding of molecules and improve the comprehension of advanced chemical and biological features. To achieve this, we prompt GPT-4o to include explanations of functional groups and their connectivity in detail. For structural descriptions, we use the annotated IUPAC name exclusively, focusing on accurately describing the structural information without emphasizing the molecular functionalities.

2. **Structure-to-Feature Relationship Explanation:** To enable a comprehensive understanding of chemical and biological features, we create structure-to-feature relationship explanations by directly relating the structural information to its molecular features. Note that learning these relationships inherently helps an LLM understand the causality between molecular structures and their features, allowing it to provide rationales when responding.

3. **Comprehensive Conversation:** Even though the two types above facilitate the learning of general knowledge about molecules, it is important to tailor LLMs to handle diverse contexts of requests. Thereby, we design conversations to cultivate the ability to handle various inquiries. Specifically, considering the hierarchical relationships of molecular features, we create comprehensive conversations by instructing to gradually deepen the level of features, progressing from structural features to chemical and biological features, to cultivate a comprehensive understanding and the step-by-step reasoning ability.

To ensure the quality of instruction-following samples, we further filter out factually incorrect ones. Inspired by LLM-as-a-judge [62], we use GPT-4o to evaluate the factual accuracy of the samples and select those with correct content, establishing 284k instruction-following samples from the

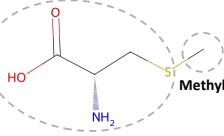

**Existing Data Example 1: Property Prediction**

Q: Is blood-brain barrier permeability (BBBP) a property of this molecule?

A: Yes

**Existing Data Example 2: Description**

S-methylcysteine is a cysteine derivative that is L-cysteine in which the hydrogen attached to the sulfur is replaced by a methyl group. It has a role as a human urinary metabolite and a plant metabolite. It is a tautomer of a S-methylcysteine zwitterion.

**Data Type 1: Detailed Structural Descriptions**

The molecule provided is a chiral amino acid that contains several identifiable substructures.

1. **Amino Group**: This functional group consists of a nitrogen atom bonded to two hydrogen atoms ($-NH_2$). It is attached to the second carbon atom ...<omitted>...

2. **Carboxylic Acid Group**: This group includes a carbon atom double-bonded to an oxygen atom and also bonded to an (-OH) group. It is attached to the end of the carbon chain, characteristic of amino acids ...<omitted>...

**Data Type 2: Structure-to-Feature Relationship Explanations**

Firstly, **the presence of the amino group (-NH2) classifies this compound as an amino acid, which contributes to its ability to act as a zwitterion**. This means that it can contain both a positive charge and a negative charge ...<omitted>...

**The sulfur atom with a methyl attachment forms a thioether. This increases the molecule's hydrophobic characteristics compared to cysteine**, which contains a more polar thiol group. ...<omitted>...

**Data Type 3: Comprehensive Conversations**

USER: Can you tell me about **the basic structure** of the given molecule?

ASSISTANT: The given molecule is a derivative of cysteine, where the hydrogen atom attached to the sulfur group is replaced by a methyl group. ...<omitted>...

USER: **How does the molecule's structure influence its chemical behavior?**

ASSISTANT: The structure of the given molecule, particularly the presence of both amino and carboxylic acid groups, contributes to its zwitterionic nature at physiological pH. ...<omitted>...

USER: **What role does this molecule play biologically?**

ASSISTANT: Biologically, S-methylcysteine serves as a metabolite in both humans and plants. ...<omitted>...

Table 2: One example of the constructed instruction data. The first two blocks show the data examples of public datasets. The following three blocks show data types and corresponding responses of GPT-4o for each data type. Entire responses of the given example are provided in Table 30, 31, and 32 of Appendix B.1

training set of the PubChem 324k dataset [25]. We refer to the constructed instruction dataset as *Mol-LLaMA-Instruct*. We note that our dataset not only aids in understanding the molecular features but also enhances explainability and reasoning capabilities by extensively addressing fundamental molecular features and various types of interactions between users and an assistant. For more details on the dataset construction, please refer to Appendix B.1.

## 3.2 Model Architecture

We now introduce the model architecture of Mol-LLaMA. Here, our goal is to accurately capture structural information, to further improve the understanding of molecular features. To this end, the model architecture consists of four components: molecular encoders, a 2D-3D blending module, a projector, and large language models.

**Molecular Encoders** We observe that each molecular encoder has distinct advantages. While the 2D encoder explicitly models the bond information and their connectivity, the 3D encoder, which represents the molecules as point clouds, captures the spatial arrangements of atoms which is crucial to understanding 3D features such as surface area and volumes. To fully leverage the strength of each encoder, we propose to use both the 2D encoder and 3D encoder. We opt MoleculeSTM [29] as the 2D encoder and UniMol [63] as the 3D encoder.

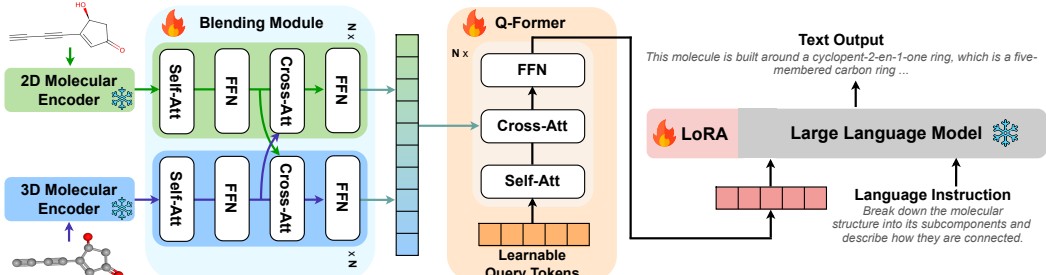

Figure 1: Illustration of the end-to-end instruction tuning stage of Mol-LLaMA. It is trained on the proposed instruction datasets, where the blending module, Q-Former, and LoRA in LLMs are trained, while the molecular encoders and LLM are frozen.

**2D-3D Blending Module** Since each representation is independently modeled, we propose a blending module that combines these molecular representations using a cross-attention scheme. Specifically, given the molecular embeddings from each encoder which are the concatenation of graph and node embeddings, self-attention and cross-attention are sequentially applied to blend the complementary information from each encoder. Then, we concatenate the 2D and 3D embeddings before forwarding to the projector, as shown in Fig. 1 (Blending Module).

**Projector** We employ Q-Former [24, 25] to project the unified molecular representations from the proposed 2D-3D blending module to an LLM, which embeds molecules by performing cross-attention between the unified molecular representations and the learnable query tokens as shown in Fig. 1 (Q-Former). It is worth noting that Q-Former is advantageous for modeling graphs, as the cross-attention guarantees the permutation invariance. We opt to initialize Q-Former with SciBERT [2].

**Large Language Models** We choose Llama-2-7b-chat [46] and Llama-3.1-8B-Instruct [16], which have demonstrated their capabilities in the multi-modal instruction tuning.

### 3.3 Training

Now, we turn out to introduce two training stages: molecular representation learning to train the projectors including the blending module and Q-Former, and end-to-end instruction-tuning.

**Molecular Representation Learning** In the first stage, we train the blending module and the Q-Former while freezing the 2D and 3D encoders. We adopt the multi-objectives to align the molecular embeddings to the molecule-relevant texts, including molecule-text contrastive learning, molecule-text matching, and molecule-grounded text generation [24, 25]. We opt to use the IUPAC name as the molecule-relevant texts instead of using descriptions. Please refer to Section B.2 for a detailed explanation of molecular representation learning.

**End-to-end Instruction Tuning** As shown in Fig. 1, we jointly train the blending module, Q-Former, and an LLM via the multi-modal instruction tuning, while freezing the 2D and 3D encoders. We instruction-tune LLMs on the proposed instruction dataset, employing LoRA [19] for the training efficiency. For the details of the instruction tuning of Mol-LLaMA, please refer Section B.2.

## 4 Experimental Results

We evaluate the capabilities of Mol-LLaMA by assessing the quality of generated responses to general questions and conducting the molecular property prediction and the molecular comprehension tasks.

### 4.1 Evaluation of General Understanding of Molecules

#### 4.1.1 Qualitative Evaluation

To show how well Mol-LLaMA understands the molecular structures and their properties, we ask a general question for a molecule whose properties are widely studied as shown in Table 1. Interestingly,

Table 3: Quantitative evaluation on the quality of generated responses for five criteria including helpfulness, relevance, accuracy, level of detail and overall score. We report the average of relative score (i.e. score of an LLM divided by score of GPT-4o) by running GPT-4o evaluation three times. † Molecular LLMs that are trained on Llama3.1-8B-Instruct following their official implementations.

| Models | STRUCTURAL | | | | | CHEMICAL | | | | | BIOLOGICAL | | | | |
|---|---|---|---|---|---|---|---|---|---|---|---|---|---|---|---|
| | Help. | Relev. | Acc. | Details | Overall | Help. | Relev. | Acc. | Details | Overall | Help. | Relev. | Acc. | Details | Overall |
| *Llama2-7B-Based* | | | | | | | | | | | | | | | |
| Llama2-7B-Chat | 0.312 | 0.333 | 0.207 | 0.284 | 0.279 | 0.447 | 0.437 | 0.304 | 0.415 | 0.394 | 0.436 | 0.422 | 0.335 | 0.449 | 0.405 |
| Mol-Instructions | 0.218 | 0.249 | 0.210 | 0.144 | 0.207 | 0.250 | 0.280 | 0.254 | 0.168 | 0.235 | 0.351 | 0.448 | 0.425 | 0.253 | 0.360 |
| LlasMol | 0.251 | 0.266 | 0.221 | 0.192 | 0.228 | 0.273 | 0.301 | 0.235 | 0.213 | 0.252 | 0.346 | 0.410 | 0.390 | 0.298 | 0.353 |
| 3D-MoLM | 0.550 | 0.541 | 0.426 | 0.542 | 0.507 | 0.669 | 0.666 | 0.557 | 0.661 | 0.628 | 0.836 | 0.894 | 0.855 | 0.892 | 0.862 |
| LLaMo | 0.314 | 0.396 | 0.348 | 0.206 | 0.310 | 0.359 | 0.459 | 0.447 | 0.240 | 0.361 | 0.498 | 0.734 | 0.803 | 0.340 | 0.568 |
| **Mol-LLaMA (Ours)** | **1.105** | **1.121** | **1.105** | **1.066** | **1.098** | **1.202** | **1.242** | **1.288** | **1.185** | **1.232** | **1.495** | **1.706** | **1.875** | **1.468** | **1.631** |
| *Llama3 or Llama3.1-8B-Based* | | | | | | | | | | | | | | | |
| Llama3.1-8B | 0.612 | 0.636 | 0.484 | 0.567 | 0.569 | 0.654 | 0.658 | 0.523 | 0.606 | 0.610 | 0.664 | 0.665 | 0.589 | 0.644 | 0.641 |
| Mol-Instructions | 0.257 | 0.315 | 0.282 | 0.166 | 0.253 | 0.274 | 0.359 | 0.322 | 0.179 | 0.276 | 0.392 | 0.547 | 0.555 | 0.259 | 0.423 |
| 3D-MoLM† | 0.778 | 0.800 | 0.680 | 0.759 | 0.749 | 0.882 | 0.936 | 0.838 | 0.854 | 0.875 | 1.105 | 1.272 | 1.292 | 1.145 | 1.191 |
| LLaMo† | 0.445 | 0.565 | 0.465 | 0.312 | 0.442 | 0.410 | 0.542 | 0.489 | 0.295 | 0.425 | 0.650 | 0.905 | 0.898 | 0.441 | 0.705 |
| **Mol-LLaMA (Ours)** | **1.126** | **1.145** | **1.154** | **1.090** | **1.125** | **1.224** | **1.266** | **1.302** | **1.211** | **1.251** | **1.578** | **1.840** | **2.030** | **1.528** | **1.744** |

Mol-LLaMA accurately predicts the main class of the given molecule, explains the related properties, and provides the rationales for the predicted properties by learning the general knowledge from the proposed dataset. In contrast, GPT-4o and 3D-MoLM misinterpret the key structures, failing to provide correct properties, while LLaMo and Mol-Instructions do not explain the relevant properties. For the additional qualitative results, please refer to Table 11, 12, and 13 in Appendix A.1.

### 4.1.2 Quantitative Evaluation

**Experimental Setting**   We first select 100 representative unseen molecules from PubChem by conducting the k-means clustering based on Morgan Fingerprints [38]. Then, we instruct to describe structural, chemical, or biological features, respectively. Please refer Appendix C.1 for detailed explanations of experimental settings.

**Evaluation Setting**   Inspired by Liu et al. [27], we leverage GPT-4o to measure the quality of generated responses. Specifically, after gathering the responses, we provide string representations and the original description as references for judging, and then instruct GPT-4o to assess the quality for the five criteria: helpfulness, relevance, accuracy, level of details, and the overall scores. We report the relative scores compared to GPT-4o (i.e. the score of an LLM divided by the score of GPT-4o). For detailed evaluation settings, please refer Appendix C.1.

**Results**   As shown in Table 3, relative scores of Mol-LLaMA are beyond 1 for all criteria, indicating that it is superior to GPT-4o in the understanding of general features of molecules, whereas the scores of other baselines are mostly lower than 1. On the other hand, Mol-LLaMA shows a significant performance improvement compared to the base LLMs and outperforms all baselines on the same architecture, indicating that the proposed instruction dataset and blending module effectively expand the molecular knowledge and introduce the reasoning capabilities to Mol-LLaMA.

### 4.2 Molecular Property Prediction

**Experimental Setting**   To assess the effectiveness of learning the general knowledge, we perform zero-shot evaluation on the PAMPA task [47]. The task is classifying the permeability of artificial membranes, requiring an understanding of essential molecular properties such as lipophilicity and molecular size. To evaluate the ability to handle diverse requests, we test on two additional prompting settings: 1) CoT [51] that instructs to provide rationales while answering and 2) prompting with task-specific information (w/ Task Info). Detailed evaluation settings are provided in Appendix C.2.

**Evaluation Setting**   We evaluate on three metrics: accuracy, fidelity, and helpfulness. Fidelity and helpfulness measure the quality of responses regardless of whether the final prediction is correct or not, to assess the qualifications of a practical assistant. We leverage GPT-4o to evaluate the fidelity and helpfulness and report the relative score compared to GPT-4o (i.e. the score of an LLM divided by the score of GPT-4o). Additionally, we report the ratio of the predicted labels to check whether an LLM is biased to predict the labels. Please refer to Appendix C.2 for detailed prompts.

Table 4: Zero-shot performances on PAMPA task. We highlight the best results in **bold**, except for the cases where all predicted labels are identical which are denoted as $^*$. N/A denotes the cases in which more than 20% of the responses do not follow the answer format. $^\dagger$ Molecular LLMs that are trained on Llama3.1-8B-Instruct following their official implementations.

| Models | Default | | | CoT | | | w/ Task Info. | | |
|---|---|---|---|---|---|---|---|---|---|
| | Acc. (Ratio) | Fidel. | Help. | Acc. (Ratio) | Fidel. | Help. | Acc. (Ratio) | Fidel. | Help. |
| GPT-4o | 48.65 (59.95) | - | - | 58.23 (47.42) | - | - | 47.17 (62.41) | - | - |
| *Llama2-7B-Based* | | | | | | | | | |
| Llama2-7B-Chat | 57.14 (36.12) | 0.517 | 0.508 | 57.53 (39.56) | 0.639 | 0.658 | 84.52 (0.00)$^*$ | 0.658 | 0.718 |
| Mol-Instructions | 49.63 (47.67) | 0.277 | 0.210 | 31.16 (70.02) | 0.314 | 0.270 | 38.18 (68.80) | 0.331 | 0.256 |
| LLaMo | **84.28** (0.74) | 0.242 | 0.187 | 84.52 (0.00)$^*$ | 0.246 | 0.191 | N/A | 0.226 | 0.185 |
| **Mol-LLaMA (Ours)** | 75.68 (11.30) | **0.781** | **0.820** | **79.61** (6.88) | **0.759** | **0.793** | 67.90 (28.75) | **0.757** | **0.744** |
| *Llama3 or Llama3.1-8B-Based* | | | | | | | | | |
| Llama3.1-8B | 56.51 (45.70) | 0.629 | 0.554 | 46.19 (58.48) | 0.795 | 0.786 | 63.64 (34.15) | 0.850 | 0.875 |
| Mol-Instructions | 55.91 (38.33) | 0.245 | 0.207 | 33.50 (73.96) | 0.299 | 0.247 | 70.47 (25.55) | 0.245 | 0.206 |
| 3D-MoLM$^\dagger$ | 46.93 (58.72) | 0.668 | 0.651 | 50.00 (51.35) | 0.671 | 0.649 | 64.86 (35.87) | 0.767 | 0.744 |
| LLaMo$^\dagger$ | 49.25 (51.74) | 0.265 | 0.212 | **64.37** (28.50) | 0.254 | 0.209 | 48.51 (53.73) | 0.401 | 0.327 |
| **Mol-LLaMA (Ours)** | **63.55** (36.86) | **0.804** | **0.829** | **64.37** (31.94) | **0.819** | **0.848** | 72.48 (17.44) | **0.927** | **0.966** |

**Result** Table 4 shows that Mol-LLaMA achieves high accuracy outperforming GPT-4o, while showing high fidelity and helpfulness scores, demonstrating that it is able to accurately predict the molecular property with helpful explanations. Further, compared to the base LLMs, Mol-LLaMA shows a performance gain for both the accuracy and the quality of the responses by learning the general knowledge and reasoning ability from our constructed dataset. In contrast, other baselines lose the ability to provide relevant and helpful explanations, showing low fidelity and helpfulness scores compared to the base LLMs. Notably, Mol-LLaMA trained on Llama3.1 shows a consistent performance improvement when using CoT prompting and providing task-specific information, showing its ability to handle diverse requests from users. We note that the models whose relative scores of fidelity and helpfulness are below 0.3 fail to provide helpful explanations, as shown in Table 26 and 27. For detailed analysis of the generated responses, please refer to Section A.8. We further provide experimental results on another task (i.e. BBBP) in Table 10 of Section A.2.

## 4.3 Molecular Comprehension Benchmark: MoleculeQA

**Experimental Settings** To evaluate the factual accuracy in the molecular comprehension, we employ MoleculeQA benchmark [33]. After fine-tuning the molecular LLMs, we report the accuracy of the answers to the questions. Please refer to Section C.3 for the introduction of MoleculeQA and the experimental details.

**Result** As shown in Table 5, Mol-LLaMA consistently outperforms the other baselines on the same model architectures on all aspects, including structure, source, property, and application. Notably, Mol-LLaMA trained on Llama-3.1-8B architecture shows the best performance, indicating that the pre-trained knowledge of Mol-LLaMA and the high expressiveness from the proposed blending module contribute to comprehending the molecular features and their behaviors. We provide entire results in Table 19 and additional experimental results for task transfer on Mol-Instructions [12] and 3D-MoIT [25] in Table 17 and 18 Section A.6.

Table 5: Performance comparison on MoleculeQA. Baseline results are taken from Lu et al. [33].

| Model | Struct. | Source | Prop. | App. | Totals |
|---|---|---|---|---|---|
| *T5-based* | | | | | |
| MolT5-small | 49.59 | 64.18 | 46.51 | 40.90 | 51.69 |
| MolT5-base | 58.01 | 65.85 | 45.14 | 42.24 | 55.39 |
| BioT5-base | 65.98 | 69.24 | 49.11 | 40.73 | 62.03 |
| MoMu-small | 52.71 | 63.44 | 44.87 | 40.57 | 52.96 |
| MoMu-base | 61.58 | 65.30 | 43.78 | 43.07 | 57.43 |
| *Galactica-based* | | | | | |
| MolCA-125M | 65.54 | 67.34 | 45.77 | 40.33 | 60.30 |
| MolCA-1.3B | 71.12 | 70.98 | 47.81 | 43.17 | 64.79 |
| *Llama2-7B-Based* | | | | | |
| Mol-Instruction | 37.46 | 47.36 | 32.69 | 29.88 | 38.37 |
| BioMedGPT-LM | 54.19 | 60.01 | 38.85 | 40.90 | 52.23 |
| 3D-MoLM | 69.64 | 68.29 | 43.19 | 43.81 | 63.31 |
| LLaMo | 65.43 | 67.14 | 45.12 | 44.33 | 61.08 |
| Mol-LLaMA | **75.33** | **73.20** | **45.26** | **45.71** | **67.97** |
| *Llama3 or Llama3.1-8B-Based* | | | | | |
| Mol-Instructions | 75.93 | 73.96 | 46.22 | 44.36 | 68.45 |
| 3D-MoLM$^\dagger$ | 76.31 | 73.64 | 47.93 | 47.33 | 69.10 |
| LLaMo$^\dagger$ | 70.56 | 66.63 | 44.60 | 45.18 | 63.74 |
| Mol-LLaMA | **77.81** | **75.50** | **49.63** | **49.30** | **70.76** |

Table 6: Ablation studies on (Left) data types and (Right) molecular representations. S denotes the detailed structural descriptions, S2F denotes the structure-to-feature relationship explanations, and Conv. denotes the comprehensive conversations. We report overall scores following the settings in Section 4.1.2. For the ablation study on data types, we report the accuracy on PAMPA task.

| | STRUCT. | CHEM. | BIO. | PAMPA | | | | STRUCT. | CHEM. | BIO. |
| Models | Overall | Overall | Overall | Default | CoT | w/ Task Info. | Models | Overall | Overall | Overall |
|---|---|---|---|---|---|---|---|---|---|---|
| S | 1.119 | 1.166 | 1.328 | 37.35 | 32.92 | 56.02 | 2D | 0.907 | 1.137 | 1.526 |
| S+S2F | 1.172 | 1.285 | 1.754 | 66.83 | 64.13 | 57.88 | 3D | 1.071 | 1.195 | 1.632 |
| Conv. | 1.166 | 0.689 | 0.887 | 41.03 | 45.95 | 83.05 | 2D+3D (Concat) | 1.037 | 1.210 | 1.741 |
| Full | 1.125 | 1.251 | 1.744 | 63.55 | 64.37 | 72.48 | 2D+3D (Blended) | 1.125 | 1.251 | 1.744 |

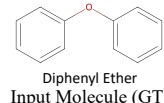

Diphenyl Ether
Input Molecule (GT)

**2D+3D (Blended)**: The given molecule, diphenyl ether, consists of two phenyl rings connected by an oxygen atom, forming an ether linkage. ... The oxygen atom is boned to both phenyl rings, making it an ether. This structure results in a symmetrical molecule with a planar geometry around the ether linkage.

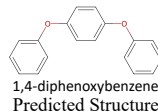

1,4-diphenoxybenzene
Predicted Structure

**2D+3D (Concat)**: The given molecule, 1,4-diphenoxybenzene, consists of a benzene ring substituted with two phenoxy groups. Each phenoxy group is attached to the benzene ring at opposite positions, specifically at the 1 and 4 positions, which are para to each other. This means that the two phenoxy groups are located on the benzene ring in a symmetrical manner. Creating a para-substituted structure.

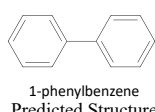

1-phenylbenzene
Predicted Structure

**3D**: The given molecule, 1-phenylbenzene, is a type of aromatic compound known as a benzene derivative. Its structure consists of two benzene rings connected by a single covalent bond. ... The presence of two benzene rings makes the molecule relatively planar, allowing for potential stacking interactions and contributing to its physical properties, such as its crystalline structure.

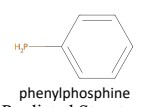

phenylphosphine
Predicted Structure

**2D**: The given molecule, phenylphosphino, consists of a phosphorus atom bonded to a phenyl group. The phenyl group is an aromatic ring, specifically a benzene ring, which is a six-carbon ring with alternating double bonds. The phosphorus atom is directly attached to the benzene ring, forming a phosphine linkage. ...

Table 7: Examples of structural descriptions for Diphenyl Ether with different molecular representation types.

## 4.4 Ablation Study

**Data Type** We ablate the different data types in Mol-LLaMA-Instruct to show the effect of each data type. As shown in Table 6 (Left), learning structural descriptions exclusively or with structure-to-feature relationship explanations helps understand the general features. On the other hand, even though learning comprehensive conversations shows an inferior understanding of chemical and biological features, it allows to handle the diverse contexts of users' requests showing a large performance gain when predicting a molecular property with task-specific information. Training on the full data balances this trade-off, showing moderate performances both on the understanding of general features and the ability to handle the users' inquiries. Please refer Table 22 of Appendix A.7 for the entire scores including helpfulness, relevance, accuracy, and level of detail.

**Blending Module** To show the effectiveness of the proposed blending module, we ablate the different types of molecular representations. As shown in Table 6 (Right), using a single type of encoder (2D and 3D) still shows a superior understanding of molecular modality compared to GPT-4o, thanks to the proposed instruction dataset. Concatenating 2D and 3D representations without the blending module (2D+3D (Concat)) results in the degeneration of the structural understanding. Using the blending module (2D+3D (Blended)) outperforms other variants, indicating that integrating the complementary information from different molecular representations is crucial to enhance the understanding of the structures and the advanced features. Qualitatively, as shown in Table 7, using a single encoder (2D or 3D) struggles to correctly predict the structural features, missing atoms or functional groups. On the other hand, while concatenating the molecular representations (2D+3D (Concat)) correctly predicts the presence of atoms and functional groups, it fails to predict the connectivity. The blending module (2D+3D (Blended)) accurately predicts the molecular structures, showing the effectiveness of integrating the complementary information from different encoders.

Table 8: Quantitative evaluation with three diverse conformations generated by RDKit and OpenBabel.

| | STRUCTURAL | | | | | CHEMICAL | | | | | BIOLOGICAL | | | | |
|---|---|---|---|---|---|---|---|---|---|---|---|---|---|---|---|
| | Help. | Relev. | Acc. | Details | Overall | Help. | Relev. | Acc. | Details | Overall | Help. | Relev. | Acc. | Details | Overall |
| Fixed | 1.127 | 1.145 | 1.154 | 1.090 | 1.125 | 1.224 | 1.266 | 1.302 | 1.211 | 1.251 | 1.578 | 1.840 | 2.030 | 1.528 | 1.744 |
| Diverse | 1.058 | 1.070 | 1.112 | 1.010 | 1.065 | 1.171 | 1.225 | 1.301 | 1.138 | 1.213 | 1.641 | 1.895 | 2.203 | 1.577 | 1.841 |

## 4.5 Further Analysis

**Robustness to Diverse Conformations** To demonstrate robustness to conformational diversity, we evaluate Mol-LLaMA on three different conformations for each representative molecule from Section 4.1.2. We note that these molecules have, on average, 11.11 rotatable bonds, ensuring diverse generated conformations. As shown in Table 8, performance with diverse conformations (Diverse) is largely consistent with that obtained using a single fixed conformation (Fixed) overall, indicating that Mol-LLaMA is robust to conformational variability.

**Few-Shot Prompting with Multiple Molecules** To evaluate the capabilities of processing and understanding multiple molecules, we apply a few-shot prompting method on the PAMPA task. As shown in Table 9, Mol-LLaMA attains a consistent performance improvement as the number of in-context examples increases, demonstrating that it effectively leverages multiple molecules and their labels. In contrast, GPT-4o fails to capitalize on additional examples; its performance declines when provided with one or three in-context examples. Across all evaluated settings, Mol-LLaMA outperforms GPT-4o, reflecting the advantages of Mol-LLaMA from domain-specialized modeling and structure-aware reasoning.

Table 9: Accuracy with few-shot prompting, which provides in-context examples of input-output pairs, on the PAMPA task.

| Models | 0-shot | 1-shot | 3-shot | 5-shot |
|---|---|---|---|---|
| GPT-4o | 48.65 | 31.45 | 46.68 | 57.74 |
| Mol-LLaMA | **63.55** | **73.46** | **77.64** | **80.10** |

## 5 Conclusion

In this work, we present Mol-LLaMA, a large molecular language model with a general understanding of molecular features and reasoning capabilities. To this end, we present a large instruction dataset by devising three data types, to cover the fundamental features from structural features to chemical and biological features and cultivate the explainability and reasoning capabilities. Further, we propose the blending module to capture the structural information from different encoder types, enhancing the understanding of structures and advanced properties. Experimentally, we show that Mol-LLaMA is capable of not only predicting molecular features accurately but also providing informative responses, implying its utility as a general-purpose assistant for molecular analysis. We hope that our dataset construction pipeline and proposed model architecture can be utilized for other modalities in the scientific fields such as proteins, RNAs, and their complexes. Further, we hope that Mol-LLaMA can be used to solve scientific problems based on its capabilities as a general-purpose molecular assistant.

**Acknowledgement** This work was supported by Institute for Information & communications Technology Planning & Evaluation(IITP) grant funded by the Korea government(MSIT) (RS-2019-II190075, Artificial Intelligence Graduate School Program(KAIST)), a grant of the Korea Machine Learning Ledger Orchestration for Drug Discovery Project(K-MELLODDY) funded by the Ministry of Health & Welfare and Ministry of Science and ICT, Republic of Korea (grant number: RS-2024-00460870), Artificial intelligence industrial convergence cluster development project funded by the Ministry of Science and ICT(MSIT, Korea) & Gwangju Metropolitan City, National Research Foundation of Korea (NRF) grant funded by the Korea government (MSIT) (No. RS-2023-00256259), Institute of Information & communications Technology Planning & Evaluation(IITP) grant funded by the Korea government(MSIT) (No.RS-2022-II220713, Meta-learning Applicable to Real-world Problems), the Institute of Information & Communications Technology Planning & Evaluation (IITP) with a grant funded by the Ministry of Science and ICT (MSIT) of the Republic of Korea in connection with the Global AI Frontier Lab International Collaborative Research (No. RS-2024-00469482 & RS-2024-00509279 & RS-2024-00509257), and Basic Science Research Program through the National Research Foundation of Korea(NRF) funded by the Ministry of Education (RS-2024-00414751).

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

# Appendix

**Organization** Appendix is organized as follows: In Section A, we provide additional results including qualitative comparison, molecular property prediction, task transfer, response analysis on PAMPA task, and ablation study. In Section B, we explain details of the instruction dataset construction and the training. In Section C, we provide experimental details for the evaluation settings. In Section D, we discuss the limitations and the societal impacts of our work.

## A  Additional Experimental Results

### A.1  Qualitative Results

In this section, we provide additional qualitative results to analyze the behaviors of Mol-LLaMA. First, we provide the entire responses of the case study in Table 1. As shown in Table 11, Mol-LLaMA understands the molecular structures and correctly predicts the main class of the given molecule (i.e. Benzodiazepine), the attached functional group (i.e. pyridine), and its biological functionalities. Interestingly, Mol-LLaMA provides additional information, detailing the effects of each structural component, such as the blood-brain barrier penetration from the pyridine ring, the binding affinity to the GABA-A receptor from the halogen atom, and the inhibitory effects of benzodiazepines. In contrast, GPT-4o, 3D-MoLM, and Mol-Instructions misinterpret the molecular structures and hence, mispredict incorrect features of molecules. On the other hand, LLaMo misinterprets the substructures (i.e. methyl group) and does not provide the distinct features of the given molecule.

Additionally, as shown in Table 12, one notable ability of Mol-LLaMA is that it provides detailed rationales for the predicted properties. Specifically, Mol-LLaMA explains the key structure (i.e., macrolides) and additional structural features (i.e., polyene chain), and then explains that these structural features are related to the specific biological properties (i.e., antifungal activity), providing principles of the predicted properties (i.e., binding to ergosterol) [44]. However, due to its limitation in interpreting raw string representations, GPT-4o misinterprets molecular structures and consequently predicts unrelated properties. 3D-MoLM mispredicts the molecular properties, while LLaMo provide the source of this molecule, which is irrelevant to the user's question.

Further, as shown in Table 13, beyond the properties annotated in PubChem, Mol-LLaMA understands various aspects of the molecular properties that can be found in the scientific literature, such as regulating the secretion of insulin [18], inhibiting the immune cells [49], and protecting neurons [9]. Similarly, GPT-4o provides relevant properties, correctly predicting the molecule name. However, LLaMo does not faithfully address the query, providing a simple treatment, not the biological functionalities.

### A.2  Molecular Property Prediction on BBBP

We additionally conduct an experiment on BBBP task, where the task is to predict whether a given molecule can penetrate the blood-brain barrier. The brain-blood barrier is a system with high selectivity by complicated transport routes [54], requiring external knowledge such as the structure of tight junctions, the interactions with membrane proteins, and the environment of BBB. We follow the experimental settings in Section 4.1.2, using three types of prompt strategies: default setting, CoT prompting, and providing task-specific information. As shown in Table 10, Mol-LLaMA shows high performance, achieving a performance improvement compared to the base LLMs. However, GPT-4o outperforms all baselines due to its extensive knowledge.

Table 10: Zero-shot performances on BBBP task. We report accuracy for the test set following the splitting from Zhou et al. [63]. The best results are highlighted in **bold**, and the second-best results are underlined.

| Models | LLM | Default | CoT | w/ Task Info. |
|---|---|---|---|---|
| GPT-4o | | **60.82** | **61.34** | **64.43** |
| Llama2 | Llama2-7B | 37.37 | 51.56 | 53.09 |
| Llama3 | Llama3.1-8B | 57.07 | 51.03 | 55.15 |
| Mol-Instructions | Llama2-7B | 52.58 | 52.58 | 51.34 |
| Mol-Instructions | Llama3-8B | - | - | 54.91 |
| LlasMol | Mistral-7B | 51.55 | 54.64 | 56.19 |
| 3D-MoLM† | Llama3.1-8B | 49.14 | 51.65 | 51.91 |
| LLaMo | Llama2-7B | - | 49.36 | 50.60 |
| LLaMo† | Llama3.1-8B | 55.44 | 55.45 | 56.91 |
| **Mol-LLaMA (Ours)** | Llama2-7B | 53.37 | 52.58 | 52.58 |
| **Mol-LLaMA (Ours)** | Llama3.1-8B | 59.54 | 55.56 | 59.08 |

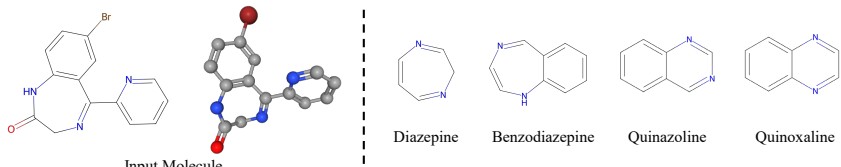

Diazepine    Benzodiazepine    Quinazoline    Quinoxaline

Input Molecule

**SMILES**: C1C(=O)NC2=C(C=C(C=C2)Br)C(=N1)C3=CC=CC=N3

**Original Description**: Bromazepam binds to the GABA receptor GABA-A, causing a conformational change and *increasing inhibitory effects of GABA*. Other neurotransmitters are not influenced. One of the *benzodiazepines* that is *used in the treatment of anxiety disorders*.

**User**: Explain the features of the given molecule that are most distinct from those of other molecules.

**Mol-LLaMA (Ours)**

The given molecule is a member of the benzodiazepine class, which is characterized by a fusion of a benzene ring and a seven-membered diazepine ring. This structure is crucial for its biological activity, particularly its role as a sedative and hypnotic agent.

One of the key features of the given molecule is the presence of a pyridine ring attached to the benzodiazepine core. This pyridine moiety is significant because it can influence the molecule's pharmacokinetic properties, such as its ability to cross the blood-brain barrier, which is essential for its function as a central nervous system depressant.

Additionally, the molecule contains a bromine atom attached to the benzodiazepine ring. The presence of this halogen atom can affect the molecule's lipophilicity and its binding affinity to the GABA-A receptor, which is the primary target of benzodiazepines. The bromine atom may also contribute to the molecule's metabolic stability and its duration of action.

The given molecule is also characterized by a ketone group within the diazepine ring, which is a common feature in benzodiazepines. This functional group is important for the interaction with the GABA-A receptor, as it plays a role in the formation of the benzodiazepine-receptor complex, enhancing the inhibitory effects of GABA.

Overall, the structural features of the given molecule, including the benzodiazepine core, the pyridine ring, the bromine atom, and the ketone group, collectively contribute to its pharmacological properties as a sedative and hypnotic agent.

**GPT-4o**

The given molecule is 6-bromo-3-(pyridin-2-yl)-1H-quinazoline-2,4-dione. Several distinctive features set this molecule apart:

1. **Quinazoline Core**: The molecule contains a quinazoline core, which is a bicyclic structure combining a benzene ring and a pyrimidine ring. This is distinct from simpler aromatic compounds due to the additional nitrogen atoms in the pyrimidine portion.

2. **Substitution Pattern**: The molecule has unique substitution on the quinazoline core:

- A bromine atom is substituted at the 6-position of the quinazoline ring. This halogenation is distinctive and can affect the molecule's reactivity and binding properties significantly, compared to non-halogenated quinazolines.

- A pyridin-2-yl group is attached at the 3-position. This makes the compound a heteroarylated quinazoline, which can modulate the electronic properties and polarity of the molecule.

3. **Presence of Two Carbonyl Groups**: The molecule has two carbonyl groups at the 2-position and 4-position of the quinazoline ring. These functional groups contribute to potential hydrogen bonding and polarity, influencing the molecule's chemical reactivity and interaction with biological targets.

4. **Heteroaromatic Ring Systems**: Besides the quinazoline core, the presence of a pyridine ring at the 3-position adds a second heteroaromatic system, increasing the overall aromatic character and introducing additional nitrogen atoms, which can serve as potential sites for interaction or coordination with metal ions.

Overall, the combination of a quinazoline structure with unique substitution patterns, including bromine halogenation and the presence of a pyridinyl group, defines the molecule's distinct chemical characteristics. This structure is particularly relevant in medicinal chemistry, where such frameworks are often explored for developing bioactive compounds.

**LLaMo**

The molecule has a benzodiazepine structure with a bromo substituent at the 7-position and a methyl group at the 1-position. It is distinct from other molecules in that it contains a bromine atom, a nitrogen atom, and a methyl group, which are not present in the other molecules.

**3D-MoLM**

The molecule with the SMILES code C1C(=O)NC2=C(C=C(C=C2)Br)C(=N1)C3=CC=CC=N3 is known as 7-bromohypnotic. It belongs to the class of compounds called quinazolines, which are characterized by a bicyclic structure consisting of a benzene ring fused to a pyrimidine ring. The molecule contains a carbonyl group (C=O) and an amide group (C(=O)N) attached to the pyrimidine ring. These functional groups contribute to the molecule's ability to form hydrogen bonds and participate in various chemical reactions. The presence of a bromine atom (Br) on the benzene ring imparts unique properties to 7-bromohypnotic. Bromine is a halogen that is more electronegative than carbon, resulting in a polar bond. This polarity can influence the molecule's reactivity and solubility in different solvents. Additionally, the molecule contains a nitrogen atom (N) in the pyrimidine ring, which can act as a basic center and participate in acid-base reactions. The presence of conjugated double bonds in the benzene and pyrimidine rings gives rise to delocalized pi-electron systems, leading to enhanced stability and potential for aromaticity. Overall, 7-bromohypnotic exhibits distinct features such as the presence of a bromine atom, carbonyl and amide groups, and conjugated double bonds, which contribute to its unique chemical and physical properties.

The molecule's structure suggests potential applications in medicinal chemistry, as quinazolines have been found to possess various biological activities, including antitumor, antimicrobial, and anti-inflammatory properties. Additionally, the presence of a bromine atom can enhance the molecule's lipophilicity, potentially affecting its bioavailability and distribution in the body. Further studies and experiments are necessary to fully understand the specific biological activities and potential therapeutic uses of 7-bromohypnotic. However, its unique molecular features make it an intriguing compound for further investigation in the field of chemistry and biology.

**Mol-Instructions**

The molecule is a quinoxaline derivative.

Table 11: Entire responses for the case study in Table 1.

**Original Description from PubChem**

Nystatin is a *polyene antifungal drug* that has broad-spectrum fungicidal and fungistatic activity against a number of yeasts and fungi, most notably Candida species. Nystatin is a *polyene macrolide antibiotic, with antifungal activity*. Upon administration, *nystatin binds to sterols in the fungal plasma membrane, thereby increasing membrane permeability. This leads to leakage of important intracellular components, cell rupture, and eventually fungal cell death.*

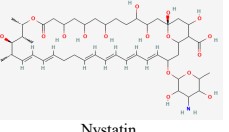

Nystatin

**User**: Could you explain the biological implication of this molecule?

---

**Mol-LLaMA**

The given molecule is a complex natural product known for its potent antifungal properties. It is produced by the fungus *Streptomyces nodosus* and is classified as a polyene macrolide. The biological implications of this molecule are primarily related to its ability to disrupt the cell membranes of fungi, making it an effective antifungal agent.

The structure of the given molecule includes a large lactone ring, which is characteristic of macrolides. This ring is conjugated with multiple conjugated double bonds, forming a polyene chain. The polyene chain is crucial for its antifungal activity, as it allows the molecule to bind to ergosterol, a key component of fungal cell membranes. This binding results in the formation of pores in the membrane, leading to the leakage of essential ions and molecules, ultimately causing cell death.

Additionally, the presence of multiple hydroxyl groups and an amino sugar moiety in the structure enhances the molecule's solubility and binding affinity to the fungal cell membrane. The carboxylic acid group contributes to the molecule's overall polarity and may play a role in its interaction with the fungal cell surface.

Overall, the given molecule's structure is intricately linked to its biological function as an antifungal agent, making it a valuable compound in the treatment of fungal infections.

---

**GPT-4o**

The molecule you've provided is Retinyl $\beta$-D-glucoside. Here's a brief explanation of its biological implications:
### Retinyl $\beta$-D-glucoside
1. **Structure and Composition**:
    - Retinyl $\beta$-D-glucoside is a derivative of vitamin A (retinol) linked to a glucose molecule.
    - The retinoid structure is vital for its function, influencing various biological processes.
2. **Biological Role**:
    - **Vision**: As a form of Vitamin A, it plays a crucial role in the visual cycle. Retinoids are essential for the formation of retinal, a molecule required for the conversion of light into visual signals in the retina.
    - **Cellular Differentiation**: Retinoids are involved in the regulation of gene expression. They influence the growth and differentiation of epithelial cells, which is critical for maintaining healthy skin and mucous membranes.
    - **Immune Function**: Vitamin A derivatives support the immune system by enhancing the mucosal barrier and facilitating the function of immune cells.
    - **Antioxidant Properties**: Retinoids can protect cells from oxidative stress, which helps to prevent chronic diseases and certain degenerative conditions.
3. **Metabolism and Function**:
    - Once ingested, retinyl $\beta$-D-glucoside would likely be metabolized to retinol and then to other active retinoids like retinal and retinoic acid to exert its physiological effects.
    - The presence of the glucose moiety may affect its absorption, distribution, and bioavailability in the body.

Overall, the biological implications of this molecule are extensive due to its role as a vitamin A derivative, impacting vision, cellular health, immune function, and more.

---

**LLaMo**

The molecule is a natural product found in Streptomyces with data available.

---

**3D-MoLM**

The molecule with the given SMILES code is a complex natural product found in Streptomyces ambofaciens. It is a polyketide compound with a highly conjugated system of double bonds, giving it a long and flexible structure. The molecule contains multiple functional groups, including hydroxyl (-OH) and carbonyl (C=O) groups, which contribute to its reactivity and potential biological activity. The molecule has a cyclic structure, with a central core consisting of a 22-membered ring. This ring contains several oxygen atoms, which are likely involved in hydrogen bonding and other interactions with biological targets. The molecule also contains a sugar moiety, specifically a 6-amino-3,4,5-trihydroxyoxane group, which is attached to the central core. This sugar group is important for the molecule's solubility and potential interactions with biological receptors. The presence of multiple hydroxyl groups suggests that this molecule may have antioxidant properties and could potentially interact with enzymes or proteins involved in oxidative stress. Additionally, the presence of an amino group indicates that this molecule may have the potential to act as a base or form hydrogen bonds with other molecules. Overall, this molecule's complex structure and functional groups suggest that it may have a wide range of biological activities, including potential antioxidant, enzyme inhibition, or receptor binding properties. Further studies are needed to fully understand its specific biological implications and potential applications.

The molecule's structure and functional groups suggest that it may have a wide range of biological activities, including potential antioxidant, enzyme inhibition, or receptor binding properties. The presence of multiple hydroxyl groups indicates that it may have antioxidant properties and could potentially interact with enzymes or proteins involved in oxidative stress. The molecule's complex structure and the presence of a sugar moiety suggest that it may have the potential to interact with biological receptors or enzymes. Further studies are needed to fully understand its specific biological implications and potential applications. The molecule's natural origin in Streptomyces ambofaciens suggests that it may have evolved to play a role in the organism's survival or defense mechanisms. Further investigation into its biosynthesis and ecological role may provide insights into its potential applications in medicine or agriculture. In summary, this molecule's complex structure and functional groups make it a promising candidate for further investigation into its biological activities and potential applications. Its natural origin and potential interactions with biological targets make it a valuable compound for further research and potential development in the fields of medicine and agriculture. Overall, this molecule's unique structure and functional groups make it a fascinating compound with potential for various biological activities. Its natural origin and potential applications make it an important target for further investigation and potential development in the fields of medicine and agriculture. Further studies are needed to fully understand its specific biological implications and potential applications.

---

Table 12: Case study for Nystatin. Mol-LLaMA correctly understands the molecular features and explains them in detail.

**Original Description from PubChem**

Adenosine is a nucleoside that is composed of adenine and d-ribose. Adenosine or adenosine derivatives play many important biological roles in addition to *being components of DNA and RNA*. For instance, adenosine *plays an important role in energy transfer* - as adenosine triphosphate (ATP) and adenosine diphosphate (ADP). It also *plays a role in signal transduction* as cyclic adenosine monophosphate, cAMP. Adenosine itself is both a *neurotransmitter* and potent *vasodilator*. When administered intravenously, *adenosine causes transient heart block* in the AV node. Because of the effects of adenosine on AV node-dependent supraventricular tachycardia, adenosine is considered a class V antiarrhythmic agent.

Adenosine

---

**User**: What are the biological functionalities of this molecule?

---

**Mol-LLaMA**

The given molecule, commonly known as adenosine, is a purine ribonucleoside composed of two main components: an adenine base and a ribose sugar. This structure allows it to play several critical biological roles:

1. **Energy Transfer and Storage**: Adenosine is a precursor to adenosine triphosphate (ATP), which is the primary energy currency of the cell. It is involved in the storage and transfer of energy in cellular processes.

2. **Signal Transduction**: As a signaling molecule, adenosine acts as a neuromodulator in the central and peripheral nervous systems. It binds to specific receptors, such as A1, A2A, A2B, and A3, influencing various physiological processes, including vasodilation, anti-inflammatory responses, and modulation of neurotransmitter release.

3. **Metabolic Regulation**: Adenosine is involved in the regulation of glucose and lipid metabolism. It affects insulin secretion and sensitivity, which is crucial for maintaining glucose homeostasis.

4. **Cardiovascular Function**: In the cardiovascular system, adenosine plays a role in regulating heart rate and blood pressure. It acts as a vasodilator, which helps in reducing blood pressure and improving blood flow.

5. **Immune System Modulation**: Adenosine can modulate immune responses by inhibiting the activity of immune cells, such as lymphocytes and macrophages. This helps in preventing excessive inflammation.

6. **Neuroprotection**: Adenosine has neuroprotective properties, which may help in protecting neurons from damage and degeneration. It is involved in the regulation of synaptic plasticity and neuronal survival.

7. **Role in DNA and RNA**: As a nucleoside, adenosine is a building block of nucleic acids, such as DNA and RNA. It is essential for the synthesis and repair of genetic material.

8. **Cellular Communication**: Adenosine is involved in cellular communication processes, such as the regulation of cell growth, differentiation, and apoptosis.

Overall, the given molecule is a multifunctional compound that plays a vital role in various physiological and biochemical processes in the body.

---

**GPT-4o**

The molecule you provided is adenosine, a nucleoside composed of adenine attached to a ribose sugar. Here are its biological functionalities:

1. **Component of Nucleic Acids**: Adenosine is a fundamental building block of RNA, where it pairs with uracil during the formation of RNA strands.

2. **Energy Transfer**: It is a part of adenosine triphosphate (ATP) and adenosine diphosphate (ADP), which are crucial molecules for energy transfer within cells.

3. **Signal Transduction**: Adenosine plays a role in cellular signaling. It acts through adenosine receptors, which are involved in various physiological processes, such as cardiovascular function, neurotransmission, and immune response.

4. **Vasodilation**: In the cardiovascular system, adenosine induces vasodilation, helping to regulate blood flow and oxygen delivery to tissues.

5. **Neurotransmission**: In the nervous system, adenosine modulates neurotransmission, playing a role in promoting sleep and suppressing arousal.

6. **Cytoprotection**: It can protect tissues from damage during hypoxia (low oxygen conditions) by modulating inflammatory responses and energy metabolism.

Understanding these roles highlights adenosine's significance in diverse biochemical and physiological processes.

---

**LLaMo**

The molecule has been used in the treatment of certain types of leukemia.

---

**3D-MoLM**

The molecule with the SMILES code C1=NC(=C2C(=N1)N(C=N2)[C@H]3[C@@H]([C@@H]([C@H](O3)CO)O)O)N is adenosine, a purine nucleoside. Adenosine is composed of a purine base, adenine, attached to a ribose sugar molecule through a $\beta$-N9-glycosidic bond. Adenosine is a crucial component of RNA and DNA, playing a vital role in various biological processes. It acts as a building block for nucleic acids, participating in DNA and RNA synthesis. Adenosine also serves as a signaling molecule, acting as a neurotransmitter and a coenzyme in cellular metabolism. The molecule's structure consists of a bicyclic purine ring system, with two nitrogen atoms at positions 1 and 9. The ribose sugar moiety is attached to the nitrogen at position 9, forming a glycosidic bond. The stereochemistry of the molecule is specified by the presence of the "@" symbol, indicating the configuration of the chiral centers. Adenosine exhibits several physical and chemical properties. It is a white, crystalline solid with a melting point of approximately 204-206°C. It is soluble in water and polar organic solvents. Adenosine is stable under normal conditions but can undergo hydrolysis in the presence of certain enzymes. In summary, adenosine is a biologically significant molecule with diverse roles in nucleic acid synthesis, signaling, and cellular metabolism. Its unique structure and properties make it an essential component of life processes in living organisms.

The raw description lacks important details and fails to provide a clear explanation of the molecule's properties. It merely states that the molecule is adenosine without elaborating on its significance, structure, or functions. Therefore, a revised description is necessary to accurately convey the molecule's characteristics and importance in biological systems.

---

Table 13: Case study for adenosine. Mol-LLaMA understands the name of molecules and their corresponding properties, providing an in-depth explanation. We highlight the properties that are not annotated in PubChem in blue.

## A.3 Expert Evaluation for Mol-LLaMA-Instruct

We assess the reliability of the constructed instruction dataset by conducting an expert evaluation. Specifically, nine experts specializing in biology, chemistry, and pharmacology are asked to rate the accuracy of fifteen detailed structural descriptions on a 4-point scale. As shown in Table 14, both coefficients are close to 1 and associated with low $p$-values, indicating a strong and statistically significant agreement between expert evaluations and GPT-4o's assessments. Notably, the average expert score for the subset of data that received a score of 4 from GPT-4o is 3.71, further demonstrating the reliability of the constructed Mol-LLaMA-Instruct.

Table 14: Pearson and Spearman correlation coefficients between accuracy assessed by GPT-4o and experts.

| | Coeff. | $p$-value |
|---|---|---|
| Pearson | 0.96 | $2.68 \times 10^{-8}$ |
| Spearman | 0.91 | $3.43 \times 10^{-6}$ |

## A.4 Performance Comparison on Diverse Evaluators

To robustly evaluate the quality of generated responses, we report the average of all metrics measured by four different evaluator LLMs. As shown in Table 15, Mol-LLaMA continues to outperform all baselines, including GPT-4o, and the overall trend remains consistent with Table 3, indicating that the evaluation of other LLMs agree with the one of GPT-4o.

Table 15: Quantitative evaluation on the quality of generated responses measured by four evaluator LLMs: GPT-4o, Qwen3-14B, Gemma-3-12B-IT, and Llama-3.1-8B-Instruct.

| Models | STRUCTURAL | | | | | CHEMICAL | | | | | BIOLOGICAL | | | | |
|---|---|---|---|---|---|---|---|---|---|---|---|---|---|---|---|
| | Help. | Relev. | Acc. | Details | Overall | Help. | Relev. | Acc. | Details | Overall | Help. | Relev. | Acc. | Details | Overall |
| *Llama2-7B-Based* | | | | | | | | | | | | | | | |
| Llama-2-7B-Chat | 0.372 | 0.374 | 0.264 | 0.359 | 0.328 | 0.534 | 0.520 | 0.405 | 0.551 | 0.492 | 0.479 | 0.408 | 0.345 | 0.598 | 0.441 |
| Mol-Instructions | 0.217 | 0.239 | 0.252 | 0.131 | 0.202 | 0.242 | 0.274 | 0.301 | 0.145 | 0.229 | 0.319 | 0.376 | 0.416 | 0.201 | 0.315 |
| LlasMol | 0.278 | 0.306 | 0.304 | 0.204 | 0.265 | 0.295 | 0.335 | 0.312 | 0.215 | 0.280 | 0.332 | 0.378 | 0.443 | 0.260 | 0.341 |
| 3D-MoLM† | 0.674 | 0.658 | 0.564 | 0.719 | 0.643 | 0.784 | 0.788 | 0.701 | 0.798 | 0.762 | 0.898 | 0.945 | 0.953 | 0.911 | 0.923 |
| LLaMo† | 0.306 | 0.407 | 0.443 | 0.179 | 0.312 | 0.345 | 0.464 | 0.524 | 0.197 | 0.356 | 0.454 | 0.630 | 0.802 | 0.236 | 0.490 |
| Mol-LLaMA | **1.042** | **1.063** | **1.063** | **0.960** | **1.038** | **1.124** | **1.179** | **1.212** | **1.044** | **1.146** | **1.376** | **1.607** | **1.683** | **1.167** | **1.468** |
| *Llama3 or Llama3.1-8B-Based* | | | | | | | | | | | | | | | |
| Llama3.1-8B | 0.669 | 0.680 | 0.560 | 0.648 | 0.634 | 0.698 | 0.693 | 0.600 | 0.695 | 0.667 | 0.705 | 0.663 | 0.617 | 0.732 | 0.674 |
| Mol-Instructions | 0.261 | 0.334 | 0.369 | 0.152 | 0.263 | 0.265 | 0.350 | 0.396 | 0.149 | 0.269 | 0.358 | 0.468 | 0.577 | 0.191 | 0.367 |
| 3D-MoLM | 0.835 | 0.874 | 0.794 | 0.782 | 0.817 | 0.902 | 0.973 | 0.902 | 0.829 | 0.900 | 1.114 | 1.295 | 1.346 | 0.970 | 1.180 |
| LLaMo | 0.480 | 0.616 | 0.560 | 0.301 | 0.473 | 0.401 | 0.534 | 0.560 | 0.245 | 0.414 | 0.596 | 0.788 | 0.834 | 0.311 | 0.603 |
| Mol-LLaMA | **1.052** | **1.081** | **1.092** | **0.964** | **1.049** | **1.139** | **1.192** | **1.225** | **1.068** | **1.163** | **1.459** | **1.699** | **1.801** | **1.210** | **1.555** |

## A.5 Temperature Sampling

Instead of the greedy sampling, we apply the temperature sampling with 0.5 temperature and report the averaged metrics of three runs in Table 16 using the four evaluator LLMs: GPT-4o, Qwne3-14B, Gemma-3-12B-IT, and Llama-3.1-8B-Instruct. Performance is maintained and Mol-LLaMA consistently outperforms the baselines, including GPT-4o, indicating that Mol-LLaMA robustly and generally shows superior performance.

Table 16: Quantitative evaluation with temperature sampling for three runs.

| Models | STRUCTURAL | | | | | CHEMICAL | | | | | BIOLOGICAL | | | | |
|---|---|---|---|---|---|---|---|---|---|---|---|---|---|---|---|
| | Help. | Relev. | Acc. | Details | Overall | Help. | Relev. | Acc. | Details | Overall | Help. | Relev. | Acc. | Details | Overall |
| Llama-3.1-8B-Instruct | 0.650 | 0.671 | 0.555 | 0.631 | 0.623 | 0.699 | 0.691 | 0.619 | 0.719 | 0.675 | 0.724 | 0.704 | 0.657 | 0.761 | 0.706 |
| Mol-Instructions | 0.258 | 0.327 | 0.355 | 0.149 | 0.258 | 0.275 | 0.345 | 0.391 | 0.153 | 0.275 | 0.355 | 0.456 | 0.583 | 0.190 | 0.370 |
| 3D-MoLM | 0.694 | 0.679 | 0.575 | 0.734 | 0.663 | 0.756 | 0.761 | 0.665 | 0.771 | 0.731 | 0.869 | 0.886 | 0.852 | 0.874 | 0.870 |
| LLaMo | 0.475 | 0.592 | 0.543 | 0.306 | 0.465 | 0.405 | 0.526 | 0.554 | 0.241 | 0.413 | 0.595 | 0.746 | 0.795 | 0.311 | 0.588 |
| Mol-LLaMA | **1.035** | **1.046** | **1.068** | **0.993** | **1.040** | **1.118** | **1.164** | **1.202** | **1.066** | **1.140** | **1.390** | **1.588** | **1.693** | **1.206** | **1.483** |

Table 17: Task transfer on molecule captioning and property prediction tasks from Mol-Instructions [12]. The baseline results are taken from Fang et al. [12] and Park et al. [41].

| | Molecule Captioning | | | | | | Property Pred. |
|---|---|---|---|---|---|---|---|
| Models | BLUE-2 | BLUE-4 | ROUGE-1 | ROUGE-2 | ROUGE-L | METEOR | MAE |
| Alpaca | 0.068 | 0.014 | 0.178 | 0.041 | 0.136 | 0.107 | 322.109 |
| Baize | 0.064 | 0.015 | 0.189 | 0.053 | 0.148 | 0.106 | 261.343 |
| LLaMA-2 | 0.059 | 0.014 | 0.164 | 0.066 | 0.148 | 0.184 | 5.553 |
| Vicuna | 0.052 | 0.011 | 0.151 | 0.055 | 0.130 | 0.168 | 860.051 |
| Galatica | 0.024 | 0.008 | 0.074 | 0.015 | 0.063 | 0.065 | 0.568 |
| Mol-Instructions (LLaMA-2) | 0.217 | 0.143 | 0.337 | 0.196 | 0.291 | 0.254 | 0.013 |
| Mol-Instructions (LLaMA-3) | 0.419 | 0.361 | 0.719 | 0.646 | 0.709 | 0.637 | 15.059 |
| LLaMo | - | 0.389 | - | - | - | 0.671 | 0.006 |
| GPT-3.5 | - | 0.022 | - | - | - | 0.197 | 0.075 |
| GPT-3.5 (ICL) | - | 0.284 | - | - | - | 0.561 | 0.075 |
| GPT-4 | - | 0.008 | - | - | - | 0.167 | 0.075 |
| GPT-4 (ICL) | - | 0.270 | - | - | - | 0.522 | 0.075 |
| **Mol-LLaMA (LLaMA-2)** | **0.478** | 0.425 | 0.761 | 0.698 | 0.750 | 0.701 | **0.0035** |
| **Mol-LLaMA (LLaMA-3)** | 0.476 | **0.426** | **0.767** | **0.708** | **0.759** | **0.707** | 0.0039 |

Table 18: Experimental results on computed property QA from 3D-MoIT benchmark [25]. We report the MAE with a valid rate (%). Baseline results are taken from Li et al. [25].

| | Basic Properties | | | | Quantum Properties | | | |
|---|---|---|---|---|---|---|---|---|
| | Weight (g/mol) | LogP | TPSA (Å$^2$) | Complexity | HOMO (eV) | LUMO (eV) | H-L Gap (eV) | SCF ($10^4$eV) |
| Uni-Mol | 20.35 | 0.59 | 13.48 | 57.24 | 0.32 | 0.35 | 0.21 | 0.45 |
| Llama2-7B | 22.10 (96%) | 1.45 (95%) | 15.87 (92%) | 69.74 (93%) | 1.24 (96%) | 1.04 (95%) | 0.88 (92%) | 0.70 (99%) |
| 2D-MoLM | 21.48 (94%) | 0.88 (96%) | 13.52 (92%) | 55.74 (94%) | 0.92 (98%) | 0.80 (96%) | 0.67 (93%) | 0.71 (99%) |
| 3D-MoLM† | 16.18 (96%) | 0.95 (96%) | 10.26 (94%) | 49.15 (95%) | 0.45 (98%) | 0.36 (96%) | 0.41 (94%) | 0.39 (99%) |
| 3D-MoLM | 14.79 (95%) | 0.66 (97%) | 9.71 (93%) | 44.85 (94%) | 0.26 (97%) | 0.25 (94%) | 0.28 (94%) | 0.35 (99%) |
| Mol-LLaMA (LLaMA-2) | 14.77 (100%) | **0.45** (100%) | 6.85 (100%) | 31.79 (100%) | **0.12** (100%) | **0.12** (100%) | **0.13** (100%) | **0.04** (99%) |
| **Mol-LLaMA (LLaMA-3)** | **14.68** (100%) | **0.45** (100%) | **6.63** (100%) | 32.25 (100%) | 0.13 (100%) | 0.13 (100%) | 0.14 (100%) | **0.04** (99%) |

Table 19: Performance comparison on MoleculeQA for general LLMs (Left) and molecular LLMs (Right). Baseline results are taken from Lu et al. [33].

| Model | Struct. | Source | Prop. | App. | Totals |
|---|---|---|---|---|---|
| *General LLM with fine-tuning* | | | | | |
| T5-small | 55.51 | 64.41 | 45.42 | 38.56 | 54.55 |
| T5-base | 60.42 | 66.42 | 45.83 | 43.74 | 58.24 |
| OPT-125M | 38.58 | 55.92 | 41.04 | 28.73 | 42.93 |
| OPT-350M | 44.39 | 60.83 | 46.24 | 40.57 | 48.05 |
| GALACTICA-6.7B | 32.35 | 41.92 | 31.05 | 28.21 | 33.96 |
| BLOOM-7.1B | 35.01 | 47.51 | 31.46 | 33.56 | 37.31 |
| Pythia-6.9B | 42.79 | 58.90 | 38.58 | 39.07 | 45.61 |
| Llama-2-7B-chat | 28.75 | 39.84 | 31.33 | 27.71 | 31.54 |
| Llama-2-13B-chat | 34.37 | 43.86 | 31.05 | 29.72 | 35.67 |
| Vicuna-v1.5-7B | 34.89 | 44.15 | 34.20 | 31.55 | 36.61 |
| Vicuna-v1.5-13B | 37.01 | 43.19 | 30.64 | 31.55 | 37.07 |
| *Large-scale Universal Models on 10-shot evaluation* | | | | | |
| Mixtral-8×7B-Instruct-v0.1 | 23.32 | 31.87 | 32.89 | 29.96 | 27.79 |
| GPT-3.5-1106-turbo | 25.60 | 37.60 | 28.04 | 32.22 | 29.29 |
| GPT-4-1106-preview | 60.94 | 50.19 | 35.57 | 43.91 | 53.47 |

| Model | Struct. | Source | Prop. | App. | Totals |
|---|---|---|---|---|---|
| *T5-based* | | | | | |
| MolT5-small | 49.59 | 64.18 | 46.51 | 40.90 | 51.69 |
| MolT5-base | 58.01 | 65.85 | 45.14 | 42.24 | 55.39 |
| BioT5-base | 65.98 | 69.24 | 49.11 | 40.73 | 62.03 |
| MoMu-small | 52.71 | 63.44 | 44.87 | 40.57 | 52.96 |
| MoMu-base | 61.58 | 65.30 | 43.78 | 43.07 | 57.43 |
| *Galactica-based* | | | | | |
| MolCA-125M | 65.54 | 67.34 | 45.77 | 40.33 | 60.30 |
| MolCA-1.3B | 71.12 | 70.98 | 47.81 | 43.17 | 64.79 |
| *Llama2-7B-Based* | | | | | |
| Mol-Instruction | 37.46 | 47.36 | 32.69 | 29.88 | 38.37 |
| BioMedGPT-LM | 54.19 | 60.01 | 38.85 | 40.90 | 52.23 |
| 3D-MoLM | 69.64 | 68.29 | 43.19 | 43.81 | 63.31 |
| LLaMo | 65.43 | 67.14 | 45.12 | 44.33 | 61.08 |
| Mol-LLaMA | **75.33** | **73.20** | **45.26** | **45.71** | **67.97** |
| *Llama3 or Llama3.1-8B-Based* | | | | | |
| Mol-Instructions | 75.93 | 73.96 | 46.22 | 44.36 | 68.45 |
| 3D-MoLM† | 76.31 | 73.64 | 47.93 | 47.33 | 69.10 |
| LLaMo† | 70.56 | 66.63 | 44.60 | 45.18 | 63.74 |
| Mol-LLaMA | **77.81** | **75.50** | **49.63** | **49.30** | **70.76** |

## A.6 Additional Results on Task Transfer

To show the effectiveness of Mol-LLaMA on the task-specific fine-tuning, we provide experimental results on the molecular captioning and molecular property prediction from Mol-Instructions in Table 17 and subsets of basic properties and quantum properties from 3D-MoIT [25] in Table 18. Mol-LLaMA exhibits the best performance on diverse tasks including molecule captioning and property prediction due to the wide-ranging knowledge about molecules. Notably, Mol-LLaMA outperforms the molecular LLMs that rely on one type of molecular encoder, indicating that incorporating complementary information from different molecular encoders is crucial to enhance adaptation to specific tasks. Additionally, we provide the entire results on MoleculeQA benchmark [33] in Table 19. Mol-LLaMA shows the best performance compared to the general LLMs and GPT-series, demonstrating its effectiveness in the task transfer along with the advantages of being open source.

## A.7 Additional Ablation Study

We further conduct an ablation study on the task transfer scenario using MoleculeQA. As shown in Table 20 (Top), each data type contributes incrementally to performance with steady improvements observed across data types (i.e., S → S+F → Full). Table 20 (Bottom) demonstrates that incorporating different molecular representations and using the blending module improves performance in total, suggesting that combining 2D and 3D representations is beneficial and that the blending module effectively integrates complementary information. Overall, the impact of data types is greater than that of model architecture, highlighting the importance of instruction dataset quality in task transfer scenarios.

Table 20: Ablation study for the data types (Top) and the blending module (Bottom) on MoleculeQA.

| Model | Struct. | Source | Prop. | App. | Totals |
|---|---|---|---|---|---|
| *Data Types* | | | | | |
| S | 72.74 | 71.41 | 49.36 | 47.27 | 66.84 |
| S+F | 76.33 | 72.19 | 46.31 | 45.91 | 68.42 |
| Conv. | 72.53 | 71.41 | 48.88 | 45.46 | 66.48 |
| Full. | 77.81 | 75.50 | 49.63 | 49.30 | 70.76 |
| *Blending Module* | | | | | |
| 2D | 72.26 | 73.79 | 49.59 | 47.75 | 67.21 |
| 3D | 76.20 | 74.42 | 50.94 | 50.45 | 69.93 |
| 2D+3D (Concat) | 76.77 | 75.61 | 49.56 | 48.58 | 70.14 |
| 2D+3D (Blended) | 77.81 | 75.50 | 49.63 | 49.30 | 70.76 |

Further, in Table 21 and 22, we report all scores of the ablation study in Table 6, including helpfulness, relevance, accuracy, and level of detail.

Table 21: Ablation studies on molecular representations. We report all scores including helpfulness, relevance, accuracy, level of detail, and overall score following the settings in Section 4.1.2.

| Models | STRUCTURAL | | | | | CHEMICAL | | | | | BIOLOGICAL | | | | |
|---|---|---|---|---|---|---|---|---|---|---|---|---|---|---|---|
| | Help. | Relev. | Acc. | Details | Overall | Help. | Relev. | Acc. | Details | Overall | Help. | Relev. | Acc. | Details | Overall |
| 2D | 0.916 | 0.968 | 0.930 | 0.835 | 0.907 | 1.129 | 1.168 | 1.163 | 1.100 | 1.137 | 1.432 | 1.631 | 1.760 | 1.362 | 1.526 |
| 3D | 1.078 | 1.085 | 1.069 | 1.047 | 1.071 | 1.185 | 1.206 | 1.237 | 1.157 | 1.195 | 1.507 | 1.713 | 1.853 | 1.443 | 1.632 |
| 2D+3D (Concat) | 1.039 | 1.076 | 1.060 | 0.980 | 1.037 | 1.187 | 1.221 | 1.256 | 1.156 | 1.210 | 1.571 | 1.827 | 2.002 | 1.537 | 1.741 |
| 2D+3D (Unified) | 1.126 | 1.145 | 1.154 | 1.090 | 1.125 | 1.224 | 1.266 | 1.302 | 1.211 | 1.251 | 1.578 | 1.840 | 2.030 | 1.528 | 1.744 |

Table 22: Ablation studies on data types. We report all scores including helpfulness relevance, accuracy, level of detail, and overall score following the settings in Section 4.1.2.

| Models | STRUCTURAL | | | | | CHEMICAL | | | | | BIOLOGICAL | | | | |
|---|---|---|---|---|---|---|---|---|---|---|---|---|---|---|---|
| | Help. | Relev. | Acc. | Details | Overall | Help. | Relev. | Acc. | Details | Overall | Help. | Relev. | Acc. | Details | Overall |
| S | 1.124 | 1.119 | 1.128 | 1.123 | 1.119 | 1.127 | 1.183 | 1.189 | 1.133 | 1.166 | 1.201 | 1.320 | 1.499 | 1.310 | 1.328 |
| S+S2F | 1.176 | 1.163 | 1.189 | 1.184 | 1.172 | 1.249 | 1.286 | 1.325 | 1.259 | 1.285 | 1.604 | 1.814 | 2.005 | 1.608 | 1.754 |
| Conv. | 1.169 | 1.152 | 1.172 | 1.176 | 1.166 | 0.674 | 0.814 | 0.787 | 0.519 | 0.689 | 0.861 | 1.079 | 1.067 | 0.589 | 0.887 |
| Full | 1.126 | 1.145 | 1.154 | 1.090 | 1.125 | 1.224 | 1.266 | 1.302 | 1.211 | 1.251 | 1.578 | 1.840 | 2.030 | 1.528 | 1.744 |

## A.8 Analysis of Generated Responses on PAMPA Task

To further understand the behaviors of LLMs and molecular LLMs, we analyze the entire responses on PAMPA task for a molecule of Fig. 2. As shown in Table 23, in the default setting, Mol-LLaMA tends to reason in a substructure-wise manner. With the chain-of-thought prompting, the response is similar to the one of the default setting as Mol-LLaMA already performs the chain-of-thought reasoning in the default setting. For the case with the task-specific information, Mol-LLaMA successfully follows the instructions, providing detailed explanations for each given property. On the other hand, as shown in Table 24, GPT-4o tends to reason in a property-wise manner, while the response with chain-of-thought prompting is not largely changed similarly to Mol-LLaMA. In contrast, as shown in Table 25, 3D-MoLM often misinterprets the molecular structures, leading to an incorrect prediction. LLaMo usually does not provide a helpful response, directly making the final decision, as shown in Table 26. As shown in Table 27, Mol-Instructions tends to generate SELFIES, not providing explanations to the queries.

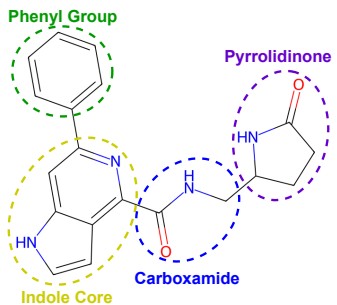

Figure 2: Input molecule

**Mol-LLaMA (Ours)**

| | |
|---|---|
| **Default** | The given molecule is a complex organic compound with several structural features that influence its permeability in the PAMPA assay. Let's break down these features and their potential impact on permeability: 
 1. **Indole Core**: The presence of an indole moiety, which is a bicyclic structure consisting of a benzene ring fused to a pyrrole ring, is significant. Indole derivatives are known for their ability to interact with biological membranes due to their planar structure and aromaticity. This can enhance the molecule's ability to permeate through the artificial membrane, contributing to high permeability. 
 2. **Phenyl Group** : The phenyl group attached to the indole core adds to the hydrophobic character of the molecule. Aromatic rings like phenyl groups can facilitate interactions with the lipid bilayer of the artificial membrane, potentially increasing permeability. 
 3. **Pyrrolidinone Moiety** : The pyrrolidinone ring, which is a five-membered lactam, introduces a degree of polarity to the molecule. This can affect the overall solubility and permeability balance. The presence of a carbonyl group within the pyrrolidinone can form hydrogen bonds, which might influence the molecule's interaction with the membrane and its permeability. 
 4. **Carboxamide Group** : The carboxamide functional group is polar and can participate in hydrogen bonding. This group can enhance the molecule's solubility in aqueous environments, which is crucial for its interaction with the PAMPA membrane. However, it may also introduce some steric hindrance, potentially reducing permeability. 
 Overall, the given molecule's structure suggests a balance between hydrophobic and hydrophilic properties. The indole and phenyl groups contribute to hydrophobic interactions, while the pyrrolidinone and carboxamide groups introduce polar characteristics. These features collectively suggest that the given molecule likely exhibits high permeability in the PAMPA assay, as it can effectively interact with the artificial membrane. Final answer: High permeability |
| **CoT** | The given molecule is a complex organic compound with several structural features that influence its permeability in the PAMPA assay. Let's break down these features and their implications: 
 1. **Indole Core**: The presence of an indole moiety, which is a bicyclic structure consisting of a benzene ring fused to a pyrrole ring, is significant. Indole derivatives are known for their ability to interact with biological membranes due to their planar structure and aromaticity. This can enhance the molecule's ability to permeate through the artificial membrane, contributing to high permeability. 
 2. **Phenyl Group**: The phenyl group attached to the indole core adds to the hydrophobic character of the molecule. Aromatic rings like phenyl groups can engage in $\pi$-$\pi$ interactions with the membrane, potentially increasing the molecule's permeability. 
 3. **Pyrrolidinone Moiety**: The pyrrolidinone ring, which is a five-membered lactam, introduces a degree of polarity to the molecule. This can affect the overall solubility and permeability balance. The presence of a carbonyl group in the pyrrolidinone can form hydrogen bonds, which might influence the molecule's interaction with the membrane and its permeability. 
 4. **Carboxamide Group** : The carboxamide functional group is polar and can form hydrogen bonds, which may enhance the molecule's solubility in aqueous environments. However, it can also affect the permeability by introducing steric hindrance and increasing the molecule's overall polarity. 
 Overall, the given molecule has a combination of hydrophobic and polar features. The indole and phenyl groups contribute to hydrophobic interactions, while the pyrrolidinone and carboxamide groups introduce polar characteristics. These features suggest that the molecule would have a balance of permeability, potentially leading to high permeability in the PAMPA assay. However, the exact permeability would depend on the specific conditions of the assay and the nature of the artificial membrane used. Final answer: High permeability. |
| **with Task Info.** | 1. **Lipophilicity**: The presence of aromatic rings, such as the phenyl group and the indolizine moiety, suggests that the given molecule is relatively lipophilic. This lipophilicity is likely to enhance its permeability through the lipid bilayer of the artificial membrane. 
 2. **Molecular Size and Weight**: The given molecule has a moderate molecular size and weight, as indicated by its classification as a pyrrolidinone. This size is not excessively large, which may allow it to pass through the membrane without significant steric hindrance. 
 3. **Polarity**: The molecule contains a pyrrolidinone group, which introduces some degree of polarity. However, the overall structure is dominated by aromatic and hydrocarbon components, suggesting that the polar surface area is not high enough to significantly impede permeability. 
 4. **Charge**: The given molecule is neutral, which is favorable for permeability as charged molecules often have reduced ability to cross lipid membranes. 
 5. **Rigidity**: The indolizine moiety contributes to the rigidity of the molecule. Rigidity can enhance membrane permeability by allowing the molecule to maintain a consistent conformation as it interacts with the membrane. 
 6. **Aromaticity**: The aromatic rings in the structure can engage in $\pi$-$\pi$ interactions with the lipid bilayer, potentially enhancing permeability. 
 7. **Hydration Energy**: The presence of the pyrrolidinone group may lower the hydration energy of the molecule, facilitating its passage through the membrane by reducing the energy required to displace water molecules. 
 8. **Membrane Affinity**: The balance of hydrophobic and hydrophilic regions in the given molecule suggests that it has a favorable membrane affinity, which is conducive to high permeability. 
 Considering these structural characteristics, the given molecule is likely to exhibit high permeability in the PAMPA assay. Final answer: High permeability. |

Table 23: Generated responses of Mol-LLaMA on PAMPA task.

| GPT-4o | |
|---|---|
| **Default** | To begin determining the permeability of the given molecule (identified as a bicyclic pyrrole compound with an embedded quinoline structure), we need to consider several molecular properties that influence permeability in the Parallel Artificial Membrane Permeability Assay (PAMPA).

Key factors that tend to influence permeability include:

1. **Lipophilicity** : Molecules with higher lipophilicity generally have higher permeability due to better membrane partitioning. The presence of aromatic rings and the overall hydrophobic character of this molecule suggests moderate to high lipophilicity.

2. **Molecular Weight** : Compounds with lower molecular weights usually have better permeability. However, this molecule appears to have a moderate molecular weight, which could suggest moderate permeability.

3. **Polarity and Hydrogen Bonding** : Molecules with several hydrogen bond donors and acceptors tend to have lower permeability due to increased interaction with water. This molecule contains multiple potential sites for hydrogen bonding (e.g., amide NH groups, ketone, and nitrogen atoms in rings), which might suggest lower permeability.

4. **Structural Rigidity and Planarity** : Rigidity and planarity can enhance membrane permeability as they may facilitate passage through lipid layers. The bicyclic and aromatic nature of the compound could potentially increase permeability.

Upon considering these factors, the multiple heteroatoms and hydrogen bonding potential in the structure suggest that, despite its potentially favorable aspects like lipophilicity, other factors potentially reduce its permeability.

Therefore, integrating these insights, the overall picture leans towards the molecule demonstrating reduced permeability. Final answer: Low-to-moderate permeability. |
| **CoT** | To evaluate the permeability of the given molecule using the Parallel Artificial Membrane Permeability Assay (PAMPA), we should consider several structural and physicochemical factors that influence passive diffusion through artificial membranes.

1. **Molecular Size**: Smaller molecules generally permeate membranes more easily than larger molecules. The given molecule is relatively large due to the presence of multiple rings and multiple atoms, indicating potential low permeability.

2. **Lipophilicity**: Lipophilicity (often estimated by LogP) plays a crucial role in membrane permeability. The presence of aromatic rings and the amide groups suggests moderate lipophilicity. The two amide groups, which can form hydrogen bonds, might reduce lipophilicity and thus permeability to lipid-like membranes.

3. **Polarity and Hydrogen Bonding**: The presence of polar functional groups such as the carbonyl and the secondary and tertiary amine groups can increase polarity and potential hydrogen bonding with the aqueous phase, decreasing permeability.

4. **Planarity**: The presence of multiple aromatic rings that might be able to stack could imply some level of planarity aiding in permeability. However, the overall complexity and potential for multiple conformations could negate this advantage.

Based on this analysis, especially considering the overall size and the presence of hydrogen-bond forming groups which can interact with the aqueous environment and hinder passive diffusion, the molecule is more likely to exhibit:

Final answer: Low-to-moderate permeability. |
| **with Task Info.** | To evaluate the PAMPA permeability of the given molecule, we need to consider several molecular properties based on the IUPAC name and SMILES string provided:

1) **Lipophilicity**: The molecule contains aromatic rings, which generally enhance lipophilicity. However, this is balanced by the presence of polar functional groups such as amides.

2) **Molecular Size and Weight**: The molecule features multiple ring structures, indicating a relatively large size and weight, which may hinder permeability.

3) **Polarity**: The presence of amide groups increases polarity, which could reduce permeability. The molecule also has a moderate potential for hydrogen bonding.

4) **Charge**: There are no apparent ionizable groups that would carry a charge at physiological pH, suggesting that the molecule remains neutral, favoring higher permeability.

5) **Rigidity**: The molecule appears to be fairly rigid due to its fused and aromatic ring structures, which can facilitate membrane permeation.

6) **Aromaticity**: The aromatic rings present will influence the hydrophobic interactions with the lipid bilayer, potentially enhancing permeability.

7) **Hydration Energy**: Polar groups and potential for hydrogen bonding can increase hydration energy, thus reducing permeability.

8) **Membrane Affinity**: The molecule's structure suggests balanced hydrophilic and hydrophobic characteristics, which could result in moderate affinity for the membrane.

Taking these properties into account, the molecule has characteristics that both enhance and reduce permeability. The moderate size, polarity, and hydration energy, combined with the presence of lipophilic and rigid structures, suggest that the molecule's permeability would not be categorized as high.

Final answer: Low-to-moderate permeability. |

Table 24: Generated responses of GPT-4o on PAMPA task.

**3D-MoLM**

| | |
|---|---|
| **Default** | The molecule with the SMILES code C1CC(=O)NC1CNC(=O)C2=C3C=CNC3=CC(=N2)C4=CC=CC=C4 is a pyrazolo[1,5-a]pyrimidine-7-carboxamide derivative. It consists of a pyrazolo[1,5-a]pyrimidine core with a carboxamide group attached at position 7. The molecule also contains a pyrrolidine ring and a phenyl ring. The presence of the carboxamide group suggests that this molecule may have the ability to form hydrogen bonds, which can influence its permeability. Additionally, the presence of the pyrrolidine ring and the phenyl ring can contribute to the overall hydrophobicity of the molecule. Based on the structure and the presence of the pyrazolo[1,5-a]pyrimidine core, it is likely that this molecule has low-to-moderate permeability to the artificial membrane. The pyrazolo[1,5-a]pyrimidine core is known to have moderate permeability, and the presence of the carboxamide group and the phenyl ring may slightly enhance or hinder the permeability, respectively. Therefore, the final answer is: Final answer: Low-to-moderate permeability. |
| **CoT** | The molecule with the SMILES code C1CC(=O)NC1CNC(=O)C2=C3C=CNC3=CC(=N2)C4=CC=CC=C4 is a pyrroloquinoline derivative. Pyrroloquinolines are a class of heterocyclic compounds that contain a pyrrole ring fused with a quinoline ring. This particular molecule is a derivative of pyrroloquinoline, with additional functional groups attached. The molecule has a pyrrolidine ring (C1CC(=O)NC1) and a pyrroloquinoline ring (CNC(=O)C2=C3C=CNC3=CC(=N2)C4=CC=CC=C4). The presence of the pyrrolidine ring suggests that the molecule may have some degree of flexibility and can adopt different conformations. The pyrroloquinoline ring is aromatic, which means it has a planar structure and exhibits delocalized pi-electrons. This aromaticity contributes to the molecule's stability and can influence its reactivity. The presence of amide groups (C(=O)N) in the molecule indicates the potential for hydrogen bonding and the ability to form intermolecular interactions. These interactions can affect the molecule's solubility and permeability. Based on the structure and functional groups present, it is likely that this molecule has low-to-moderate permeability to the artificial membrane. The presence of aromatic rings and amide groups can influence the molecule's ability to pass through the membrane, but the overall size and complexity of the molecule may limit its permeability. Therefore, the final answer is: Final answer: Low-to-moderate permeability. |
| **with Task Info.** | The molecule with the SMILES code C1CC(=O)NC1CNC(=O)C2=C3C=CNC3=CC(=N2)C4=CC=CC=C4 is a complex compound with various chemical and physical properties that influence its permeability in the Parallel Artificial Membrane Permeability Assay (PAMPA). Let's analyze these properties to determine the permeability of the molecule. 1. Lipophilicity: The presence of multiple aromatic rings, such as the phenyl ring, suggests that this molecule is lipophilic. Lipophilicity generally correlates with increased permeability, up to a certain threshold. 2. Molecular Size and Weight: The molecule has a relatively large size due to the presence of multiple rings and functional groups. This size may limit its permeability. 3. Polarity: The molecule contains polar functional groups, such as amide and carbonyl groups, which contribute to its overall polarity. However, the presence of aromatic rings and hydrophobic substituents, like the phenyl ring, suggests that the molecule may have a balanced affinity for both the aqueous phase and the lipid bilayer. 4. Charge: The molecule is neutral, which is generally favorable for permeability. 5. Rigidity: The presence of multiple rings and the rigidity of the molecule may contribute to its permeability. 6. Aromaticity: The molecule contains aromatic rings, which can influence lipophilicity and molecular interactions with the lipid bilayer, thereby affecting permeability. 7. Hydration Energy: The presence of polar functional groups may result in higher hydration energy, which could potentially hinder membrane permeation. 8. Membrane Affinity: The balanced affinity for both the aqueous phase and the lipid bilayer suggests that the molecule may exhibit better PAMPA permeability. Based on these properties, the molecule is expected to have low-to-moderate permeability to the artificial membrane. The presence of polar functional groups, large size, and potential hydration energy limitations may hinder its diffusion through the lipid bilayer. However, the lipophilicity, rigidity, and aromaticity may contribute to its permeability to some extent. Therefore, the final answer is: Final answer: Low-to-moderate permeability. |

Table 25: Generated responses of 3D-MoLM on PAMPA task.

**LLaMo**

| | |
|---|---|
| **Default** | The molecule has low-to-moderate permeability to the artificial membrane. |
| **CoT** | The molecule has high permeability to the artificial membrane. |
| **with Task Info.** | The molecule has high lipophilicity, low molecular size and weight, low polarity, and is neutral. It also has a high degree of rigidity and aromaticity, which are all factors that contribute to higher permeability. Final answer: The molecule has high permeability. |

Table 26: Generated responses of LLaMo on PAMPA task.

**Mol-Instructions**

| | |
|---|---|
| **Default** | [C][C][C][=Branch1][C][=O][N][C][Ring1][=Branch1][C][N][C][=Branch1][C][=O][C][=C][C][=C][N] [Branch1][=C][C][=C][C][=C][C][=C][Ring1][=Branch1][C][=C][Ring1][=C][N][=C][Ring1][P][C][=C][C] [=C][C][=C][Ring1][=Branch1] Final answer: High permeability |
| **CoT** | [O][=C][C][=C][C][=C][N][C][Ring1][Branch1][=C][C][=Branch1][Ring2][=N][Ring1][=Branch2][C][=C] [C][=C][C][=C][Ring1][=Branch1].[C][C][C][=Branch1][C][=O][N][C][Ring1][=Branch1][C][N] Final answer: Low-to-moderate permeability. |
| **with Task Info.** | [C][C][C][=Branch1][C][=O][N][C][Ring1][=Branch1][C][N][C][=Branch1][C][=O][C][=C][C][=C][N] [Branch1][#Branch1][C][=C][C][=N][Ring1][=Branch1][C][=Branch1][C][=O][N][Ring1][N][C][=C][C][=C] [C][=C][Ring1][=Branch1] Final answer: High permeability |

Table 27: Generated responses of Mol-Instructions on PAMPA task.

# B  Details of Mol-LLaMA

## B.1  Instruction Dataset Construction

**Details of GPT-assisted Data Generation**    First, we randomly select 100k molecules from the train set of PubChem following the splitting from Li et al. [25] for each data type including detailed structural descriptions, structure-to-chemical features, structure-to-biological features, and comprehensive conversations. Then, we employ GPT-4o-2024-08-06 to construct the instruction datasets with different prompts as shown in Table 28. For the data filtering, the prompts are provided in Table 29, where we select samples whose scores are 4. As a result, we collect 284k instruction-following samples, establishing 77k samples for the detailed structural descriptions, 147k samples for the structure-to-feature relationship explanations, and 60k samples for the comprehensive conversations. The entire responses for the example in Table 2 are provided in Table 30, 31 and 32.

---

### Prompt for Detailed Structural Description

**System**

You are a chemical assistant and you are given a molecule with the IUPAC name.

Provide a detailed explanation of the given molecule at the molecular structural level. Explain which substructures and functional groups are contained and how they are connected.

---

**User**

Input molecule (IUPAC name): `{IUPAC name}`

---

### Prompt for Structure-to-Feature Relationship Explanation

**System**

You are a {level} assistant that can analyze the {level} properties of a single molecule. A molecule is given as the IUPAC name, accompanied by a description.

Based on the provided IUPAC name and the description, explain **the `{level}` properties** in a detailed manner by relating the {level} properties to its structural information.

---

**User**

Input molecule (IUPAC name): `{IUPAC name}`

Description: `{Description}`

---

### Prompt for Comprehensive Conversation

**System**

You are an AI chemical assistant with extensive knowledge of molecular properties. You are given a molecule with the IUPAC name and its description.

Your task is to design a conversation between you (e.g. AI chemical assistant) and a user asking about this molecule. Design a sequence of pairs of questions and answers that gradually deepen the level of the conversation, from structural information and chemical properties to biological functionalities.

Include questions asking about the molecule's structural, chemical, and biological features, including functional groups, the most specific compound species name, corresponding chemical and biological properties, and functionalities, etc.

---

**User**

Input molecule (IUPAC name): `{IUPAC name}`

Description: `{Description}`

---

Table 28: Prompts for constructing the instruction dataset for each data type. `{level}` is 'chemical' or 'biological'.

**System**

You are an assistant specializing in chemistry and biology. You are provided with a molecule's IUPAC name and its {level} description.

Your task is to evaluate the factual accuracy of the given description based on the provided IUPAC name.

Assign a score from 1 to 4 based on the following criteria:

1: All contents are factually incorrect

2: Some contents are factually correct, but most are factually incorrect

3: Most contents are factually correct, but some are factually incorrect

4: All contents are factually correct

Indicate your score in the format: "Score: ...".

**User**

Input molecule (IUPAC name): {IUPAC name}

Description: {Description}

Table 29: Prompts for filtering correct samples. {level} is one of the following: 'structural', 'chemical', and 'biological'.

---

**Context Type 1: IUPAC Name**

(2R)-2-amino-3-methylsulfanylpropanoic acid

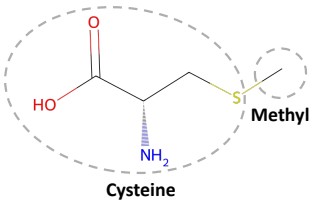

**Data Type 1: Detailed Structural Descriptions**

The molecule provided is a chiral amino acid that contains several identifiable substructures.

1. **Amino Group**: This functional group consists of a nitrogen atom bonded to two hydrogen atoms ($-NH_2$). It is attached to the second carbon atom of the main carbon chain, **which is typical for amino acids** as they feature the amino group connected to the alpha carbon.

2. **Carboxylic Acid Group**: Represented by $-COOH$, this group includes a carbon atom double-bonded to an oxygen atom and also bonded to an $-OH$ group. It is attached to the end of the carbon chain, **characteristic of amino acids** which bear a carboxylic acid group at the terminal carbon.

3. **Main Carbon Chain**: The molecule has a three-carbon chain serving as the backbone. The central carbon, referred to as the alpha carbon, is a chiral center, meaning it is bonded to four different substituents: the amino group, the carboxylic acid group, a hydrogen atom, and a side chain.

4. **Methylsulfanyl Group**: Attached to the third carbon is a side chain consisting of a sulfur atom single-bonded to a methyl group ($-SCH_3$). This group acts as the side chain substituent and characterizes the molecule as containing a **thioether linkage**, where the sulfur atom replaces the carbon in the typical ether linkage.

5. **Chirality**: The molecule is chiral, with the alpha carbon being the stereocenter. Its configuration is designated as $R$, indicating the spatial arrangement of its substituents according to the Cahn-Ingold-Prelog priority rules.

Overall, this molecule is a sulfur-containing, optically active alpha amino acid with a focus on the specific arrangement of its functional groups including the amino, carboxylic acid, and a distinct methylsulfanyl side chain. This structure is essential for understanding its interactions and functions in biological systems.

Table 30: One example of the constructed instruction data. The top block shows the example of context types when prompting GPT-4o. The following block show the responses of GPT-4o.

**Context Type 1: IUPAC Name**

(2R)-2-amino-3-methylsulfanylpropanoic acid

**Context Type 2: Description**

S-methylcysteine is a cysteine derivative that is L-cysteine in which the hydrogen attached to the sulfur is replaced by a methyl group. It has a role as a human urinary metabolite and a plant metabolite. It is a tautomer of a S-methylcysteine zwitterion.

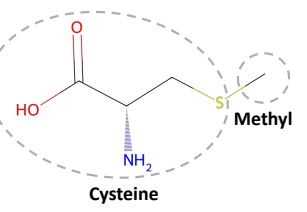

## Data Type 2: Structure-to-Feature Relationship Explanations (Chemical Features)

**The given molecule is a derivative of cysteine**, an amino acid known for containing a thiol group. In the given molecule, this thiol group has one of its hydrogen atoms replaced by a methyl group, making it a methylthioether. This structural tweak to cysteine results in distinct chemical properties.

Firstly, **the presence of the amino group (-NH2) classifies this compound as an amino acid, which contributes to its ability to act as a zwitterion.** This means that in aqueous solutions or physiological pH, it can contain both a positive charge (on the ammonium group) and a negative charge (on the carboxylate group), enhancing its solubility in water.

**The sulfur atom with a methyl attachment forms a thioether. This increases the molecule's hydrophobic characteristics compared to cysteine**, which contains a more polar thiol group. This methylsulfanyl group can also affect the molecule's reactivity and interaction with biochemical systems, possibly altering how it participates in or affects metabolic pathways.

The stereochemistry indicated by 'R' specifies the spatial arrangement of groups around the chiral carbon atom, which corresponds to natural configurations found in biological systems like human and plant metabolites. This natural configuration is crucial for its biological activity and integration into metabolic processes.

As a human urinary metabolite, the presence of this compound in urine suggests its role in human metabolism where it might play a role in sulfur metabolism or act as a detoxification product. In plants, it could partake in sulfur metabolism and possibly aid in detoxification or defense processes due to environmental stressors.

In summary, the structural modification from cysteine to include a methylthioether enhances its lipophilic properties and potentially alters its metabolic and biochemical roles due to changes in reactivity, solubility, and interaction with enzymes or other biomolecules.

## Data Type 2: Structure-to-Feature Relationship Explanations (Biological Features)

The given molecule, known as S-methylcysteine, is a derivative of the amino acid cysteine. In its structure, the core amino acid framework is maintained, which consists of an amino group, a carboxylic acid group, and a side chain. In the side chain of this molecule, the sulfur atom of cysteine is bonded to a methyl group, distinguishing it from the native structure of cysteine.

The biological properties of the given molecule are influenced significantly by this methylation at the sulfur atom. This modification preserves some aspects of cysteine's chemical behavior but also endows the molecule with unique characteristics:

1. **Human Urinary Metabolite Role**: The metabolite role of the molecule in humans suggests it participates in or results from various metabolic processes. The transformation of cysteine to S-methylcysteine may occur as part of detoxification pathways or sulfur metabolism. The presence in urine indicates the body's method of excreting or regulating sulfur-containing compounds, possibly involved in processes managing oxidative stress or sulfur balance.

2. **Plant Metabolite Role**: In plants, sulfur-containing amino acids like the given molecule can function in sulfur storage, cellular redox regulation, or even in the defense against stress. Methylation of cysteine could help modulate its reactivity, buffering against electrophile attacks or preventing unwanted formation of disulfide bonds, which is crucial for cellular homeostasis.

3. **Tautomeric Properties**: The tautomeric relationship of the given molecule with its zwitterionic form is instrumental in understanding its behavior in biological environments, such as different pH conditions. As a zwitterion, it can exhibit varied solubility and stability profiles, influencing its transport and reactivity within both human and plant systems.

The methylsulfanyl group adds to the biological properties by contributing a degree of non-polarity and slightly altering the exact steric and electronic nature of the side chain compared to cysteine, which can influence enzyme interactions, protein structure stability, and metabolic pathways where sulfur chemistry is paramount. Overall, despite being derived from cysteine, the methyl modification gives the molecule distinct roles and characteristics in living organisms.

Table 31: One example of the constructed instruction data. The top block shows the example of context types when prompting GPT-4o. The following two blocks show the responses of GPT-4o.

**Context Type 1: IUPAC Name**

(2R)-2-amino-3-methylsulfanylpropanoic acid

**Context Type 2: Description**

S-methylcysteine is a cysteine derivative that is L-cysteine in which the hydrogen attached to the sulfur is replaced by a methyl group. It has a role as a human urinary metabolite and a plant metabolite. It is a tautomer of a S-methylcysteine zwitterion.

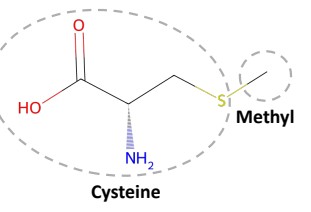

---

**Data Type 3: Comprehensive Conversations**

USER: Can you tell me about **the basic structure** of the given molecule?

ASSISTANT: Certainly! The given molecule is a derivative of cysteine, where the hydrogen atom attached to the sulfur group is replaced by a methyl group. This makes it include a sulfur-containing side chain as part of its structure. It has an amino group (-NH2), a carboxylic acid group (-COOH), and a methylsulfanyl side chain, all connected to the central carbon atom.

USER: What are **the functional groups** present in this molecule?

ASSISTANT: The molecule contains the following functional groups: an amino group (-NH2), a carboxylic acid group (-COOH), and a methylthio group (methylsulfanyl, -SCH3).

USER: What is the stereochemistry of this molecule?

ASSISTANT: The given molecule has a chiral center at the 2nd carbon atom and is specified to be in the (R)-configuration, meaning its spatial arrangement is determined relative to this stereocenter.

USER: Can you tell me more about **the specific species** of this compound?

ASSISTANT: The compound is known as S-methylcysteine. It is a specific derivative of cysteine with a distinctive methyl substitution on the sulfur atom, differing from the typical cysteine structure.

USER: How does the molecule's structure influence **its chemical behavior**?

ASSISTANT: The structure of the given molecule, particularly the presence of both amino and carboxylic acid groups, contributes to its zwitterionic nature at physiological pH. The substitution of a methyl group on the sulfur atom confers specific interactions and stability to the thiol, altering its reactivity compared to standard cysteine.

USER: **What role does this molecule play biologically?**

ASSISTANT: Biologically, S-methylcysteine serves as a metabolite in both humans and plants. It is identified as a human urinary metabolite and also plays a role as a plant metabolite, indicating its potential involvement in metabolic pathways within these organisms.

USER: What might be some of **the functions or activities attributed to this molecule**?

ASSISTANT: The molecule is involved in several functions due to its presence in metabolic processes. In plants, S-methylcysteine may play a role in sulfur transport or storage. Its function as a human metabolite could potentially affect pathways related to detoxification or protein synthesis where cysteine and its derivatives play roles.

Table 32: One example of the constructed instruction data. The top block shows the example of context types when prompting GPT-4o. The following block show the responses of GPT-4o.

**Instructions** We provide the instructions for the comprehensive conversations, the detailed structural description, the structure-to-chemical feature relationship explanations, and the structure-to-biological feature relationship explanations in Table 33, 34, 35, and 36, respectively. Instructions for each data type have similar semantics. For the comprehensive conversations, the user prompts are the generated questions.

---

**System**

You are a helpful assistant specializing in chemistry and biology. The instruction that describes a task is given, paired with molecules. Write a response that appropriately completes the request.

---

Table 33: List of instructions for the comprehensive conversations.

**System**

You are a helpful assistant specializing in chemistry and biology. The instruction that describes a task is given, paired with molecules. Provide a comprehensive response that appropriately completes the request.

**User**
- Explain the components and how they are linked within the provided molecule.
- Detail the structural parts of the molecule and their interconnections.
- Outline the individual subunits of the molecule and describe their arrangement.
- Provide an analysis of the molecular substructures and how they are bonded together.
- Identify the segments of the molecule and elaborate on their attachments.
- Break down the molecular structure into its subcomponents and describe how they are connected.
- Map out the substructures within the molecule and illustrate how they are linked.

Table 34: List of instructions for the detailed structural descriptions.

**System**

You are a helpful assistant specializing in chemistry and biology. The instruction that describes a task is given, paired with molecules. Provide a comprehensive response that appropriately completes the request.

**User**
- Provide an in-depth explanation of the chemical characteristics of the given molecule.
- Elaborate on the detailed chemical attributes and properties of the molecule.
- Describe the chemical properties of the provided molecule with comprehensive detail.
- Offer a thorough analysis of the chemical characteristics of the compound.
- Discuss the chemical properties of the given compound extensively and in detail.
- Present an in-depth overview of the chemical attributes of the provided compound.
- Explain the detailed aspects of the chemical properties of the molecule.
- Analyze the the molecule's chemical properties with an in-depth approach.
- Present a detailed report on the chemical traits of the compound.

Table 35: List of instructions for the structure-to-chemical feature relationship explanations.

**System**

You are a helpful assistant specializing in chemistry and biology. The instruction that describes a task is given, paired with molecules. Provide a comprehensive response that appropriately completes the request.

**User**
- Provide a comprehensive explanation of the biological characteristics of the given molecule, focusing on how its main substructures relate to its biological properties.
- Discuss the molecule's biological properties thoroughly, emphasizing the connection between its key substructures and their functions.
- Elaborate in detail on the biological attributes of the provided compound, explaining how its primary substructures are linked to its properties.
- Analyze the biological properties of the given compound, providing an in-depth explanation of how the core substructures of the molecule influence these properties.
- Describe the biological characteristics of the given molecule in detail, paying particular attention to how its main structural components affect its behavior.
- Offer an in-depth discussion of the biological traits of the molecule, specifically highlighting the relationship between the core parts of the molecule and its properties.
- Present a detailed analysis of the biological properties of the provided molecule, focusing on how the essential substructures within the molecule correlate with these properties.
- Give an in-depth explanation of the biological properties of the provided molecule, especially how its core substructures are associated with these properties.
- Outline the biological properties of the given compound comprehensively, emphasizing the interplay between its main substructures and its biological behavior.

Table 36: List of instructions for the structure-to-biological feature relationship explanations.

## B.2 Training Details

**Blending Module**    The number of heads of the blending module is 8, and the number of blocks is 4, where each block consists of a sequence of one self-attention block and one cross-attention block.

**Molecular Representation Learning**    In the molecular representation learning stage, Q-Former is constituted of two transformers: molecular transformer and text transformer as shown in Fig. 3. The molecular transformer embeds the molecular information by the cross-attention between learnable query tokens and the molecular embeddings with an additional cross-attention block. The text transformer models the molecule-relevant texts while maintaining the original transformer architecture.

To train Q-Former, we adopt three training objectives proposed in Li et al. [25]: molecule-text matching, molecule-text contrastive learning, and molecule-grounded text generation. Specifically, we choose IUPAC name as the molecule-relevant text to compactly learn the molecular structures. Therefore, we refer them to the structure-IUPAC matching, structure-IUPAC contrastive learning, and structure-grounded IUPAC generation. The structure-IUAPC contrastive learning and structure-IUPAC matching aim to learn the similarity via the cosine similarity or the binary classification, respectively. The structure-grounded IUPAC generation aims to learn the text generation via the next token prediction. For each training objective, the self-attention masking strategies are different. For the structure-IUPAC contrastive learning, self-attention is performed on each modality. For structure-IUPAC matching, the self-attention is performed for all tokens without masking. For the structure-grounded IUPAC generation, the causal mask is applied.

The blending module and the Q-Former are trained for 50 epochs. The optimizer is AdamW optimizer [31] with a weight decay of 0.05 and a cosine scheduler with 1000 steps of linear warmup where the peak and minimal learning rates are 1e-4 and 5e-6. The number of query tokens is 8 and the batch size is 256.

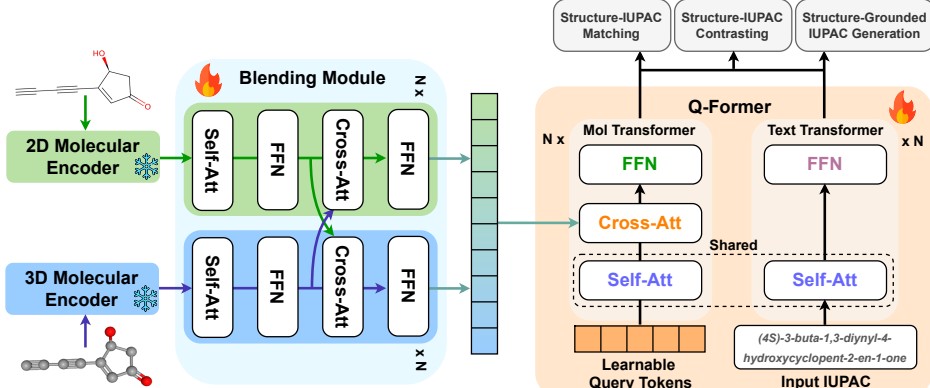

Figure 3: Detail illustration of blending module and Q-Former and their training.

**End-to-end Instruction Tuning**    We leverage LoRA [19] where the rank ($r$) is 8, $\alpha$ is 32, and the dropout ratio is 0.1. We use the same optimizer configuration in the molecular representation learning stage, while training for 10 epochs with 128 batch sizes.

**Resources**    We train Mol-LLaMA on NVIDIA H100 and NVIDIA A100 80GB.

## C    Experimental Details

### C.1    Quantitative Evaluation on General Understanding of Molecules

To evaluate the general understanding of molecules, we ask general questions about structures and chemical and biological properties, respectively, as follows: "Explain the structural features of the given molecule.", "Explain the chemical properties of the given molecule.", and "Explain the biological properties of the given molecule.". To assess the quality of responses, GPT-4o is provided the IUPAC name, the original descriptions annotated in PubChem [21], and the questions asked to the assistants as references. Then, given the responses of two assistants, GPT-4o assesses the scores of each response for five criteria as shown in Table 37.

**System**

You are a helpful assistant specializing in chemistry and biology. Your task is to evaluate the performance of two AI assistants in responding to a user question about a molecular explanation.

For your reference, the SMILES notation, IUPAC name, and a description of the given molecule are provided.

Evaluate each assistant's response based on the following criteria: helpfulness, relevance, accuracy, and level of detail. Rate each criterion on a scale of 1 to 10, where a higher score indicates better performance. Additionally, provide an overall score for each assistant's response on a scale of 1 to 10.

First output the scores of each assistant in the following format:

[Assistant n]

- Helpfulness: ...

- Relevance: ...

- Accuracy: ...

- Level of detail: ...

- Overall: ...

In the subsequent line, please provide a comprehensive explanation of your evaluation, avoiding any potential bias and ensuring that the order in which the responses were presented does not affect your judgment.

---

**User**

[Molecule Information]

SMILES: {SMILES}

IUPAC Name: {IUPAC name}

Description: {Description}

[Question]

Explain the {level} features of the given molecule.

[Assistant 1]

{Response of Assistant 1}

[End of Assistant 1]

[Assistant 2]

{Response of Assistant 2}

[End of Assistant 2]

Table 37: Prompts for evaluating responses for the general questions. {level} is one of the following: "structural", "chemical", or "biological".

## C.2 Molecular Property Prediction

We first generate the molecular conformations using RDKit and OpenBabel, then split the train, valid, and test datasets using the predefined random splitting from the TDC benchmark [47]. The prompts for predicting PAMPA results are provided in Table 38 and the prompts for evaluating the reasoning processes on the PAMPA task are provided in Table 39. If the generated responses do not follow the designated format of the final answer, we add the final answer format (i.e. "Final answer: ") at the end of the generated responses and let LLMs generate in succession to make the final decision based on their previous reasoning process. We use the greedy decoding strategy to generate the responses of LLMs.

**System**

You are a drug discovery assistant tasked with predicting the permeability of a molecule in the Parallel Artificial Membrane Permeability Assay (PAMPA). Specifically, your role is to determine whether a molecule has high permeability or low-to-moderate permeability to the artificial membrane.

Consider the following properties of molecules:

1) Lipophilicity: Higher lipophilicity generally correlates with increased permeability, up to a certain threshold.

2) Molecular Size and Weight: Smaller molecules tend to have higher permeability.

3) Polarity: Low polar surface area and low hydrogen bond donors/acceptors are associated with higher permeability.

4) Charge: Neutral molecules typically have better permeability compared to charged species, which are less likely to diffuse through the hydrophobic lipid bilayer.

5) Rigidity: A high degree of rigidity often permeate membranes more easily.

6) Aromaticity: The presence of aromatic rings can influence lipophilicity and molecular interactions with the lipid bilayer, thereby affecting permeability.

7) Hydration Energy: Lower hydration energy generally improves membrane permeation.

8) Membrane Affinity: Compounds with a balanced affinity for both the aqueous phase and the lipid bilayer tend to exhibit better PAMPA permeability.

Your final answer should be formatted as either : 'Final answer : High permeability.' or 'Low-to-moderate permeability.'

**User**

Determine the permeability of the given molecule to the artificial membrane.

Please provide a rationale for your answer.

Table 38: Prompts for PAMPA task. For the default setting, the blue prompt and green prompt are not included. For the CoT promoting, we add the blue prompt, and, for the case with the task-specific information (w/ Task Info), we add the green prompt not including the blue prompt.

---

**System**

You are a helpful assistant specializing in chemistry and biology, whose role is to evaluate the quality of the reasoning process of an AI assistant in predicting the permeability of molecules in the Parallel Artificial Membrane Permeability Assay (PAMPA).

For your reference, the SMILES of the given molecule is provided.

Evaluate the quality of each assistant's response based on the criteria below:

Fidelity: It evaluates the soundness and relevance of the reasoning process by assessing whether the reasoning is valid to appropriately address the given task.

Helpfulness: It evaluates the quality of the reasoning process by assessing whether the reasoning is clear, informative, and helpful to the user.

First, provide an explanation of your assessment, and then evaluate the score on a scale of 1 to 10, where a higher score indicates better quality. Follow the format in the below example:

Explanation of the evaluation:

Final Decision:

[Assistant n]

- Fidelity : ...

- Helpfulness : ...

**User**

[Molecule Information]

SMILES: {SMILES}

[Assistant 1]

{Response of Assistant 1}

[End of Assistant 1]

[Assistant 2]

{Response of Assistant 2}

[End of Assistant 2]

Table 39: Prompts for evaluating reasoning results in PAMPA task.

### C.3 Molecular Comprehension Benchmark: MoleculeQA

We first generate the 3D conformations of molecules using RDKit and OpenBabel. Then, we fine-tune molecular LLMs including Mol-LLaMA, 3D-MoLM, Mol-Instructdions, and LLaMo on the training dataset in Molecule QA benchmark for 20 epochs, where the total batch size is set to 256 with gradient accumulation, the learning rate is fixed to 1e-4, and the weight decay is set to 0.05 with AdamW [31] optimizer. The fine-tuned models are evaluated on the greedy decoding strategies on the test datasets.

## D  Limitations and Societal Impacts

**Limitations**    In this work, we present Mol-LLaMA, a large molecular language model that grasps broad and general knowledge of molecules. Although Mol-LLaMA has a wide-ranging understanding of molecular features, it is focused on molecular analysis of molecular properties. There might be a request to generate molecules with desired properties, which could be achieved by utilizing our model as a reward model. We leave this framework as future work.

**Societal Impacts**    We experimentally demonstrate that Mol-LLaMA is capable of not only accurately predicting the molecular properties, but also providing helpful explanations of its reasoning. We belive that our model can be further utilized to accelerate scientific discovery by efficiently predicting and understanding molecular properties, reducing the time-consuming wet-lab experiments. However, one might maliciously use our model for discovering harmful and toxic substances. We sincerely hope that our method will not be used for a bad purpose.

