# OpenReview forum: "Mol-LLaMA: Towards General Understanding of Molecules in Large Molecular Language Model"
_NeurIPS.cc/2025/Conference — NeurIPS 2025 poster_

### Official Review · Reviewer_voqH · 2025-06-16

**Clarity:** 2
**Significance:** 2
**Originality:** 2
**Rating:** 4
**Confidence:** 4

**Summary:**

This paper presents Mol-LLaMA, a general-purpose molecular assistant designed to perform molecular analysis with enhanced reasoning and explainability. The primary contribution is the creation of Mol-LLaMA-Instruct, a large instruction-following dataset generated via GPT-4o, incorporating hierarchical reasoning across structural, chemical, and biological levels. The dataset construction leverages annotated IUPAC names and curated PubChem descriptions, using GPT-4o to generate multi-level reasoning samples.

The model architecture integrates 2D and 3D molecular encoders, a 2D–3D blending module, and a Q-Former to project molecular features into a large language model space. The authors adopt a two-stage training procedure involving molecular representation learning followed by multi-modal instruction tuning. Evaluation is conducted using GPT-4o as an automated judge across multiple criteria such as helpfulness, relevance, and accuracy.

**Questions:**

How well justified is the choice of molecular encoders (MoleculeSTM for 2D, UniMol for 3D)? Were alternative encoders considered or compared?

**Ethical Concerns:**

["NO or VERY MINOR ethics concerns only"]

**Final Justification:**

While I still believe the model-side method lacks novelty, I acknowledge that the newly introduced dataset is valuable. The inclusion of preliminary human evaluation results with a fairly high R2 is promising. Given this, I am willing to raise my rating, as I believe this work could make a useful contribution to the research community.

**Limitations:**

I did not observe any major formatting issues during my review of the paper.

**Paper Formatting Concerns:**

I did not observe any major formatting issues during my review of the paper.

**Quality:**

2

**Strengths And Weaknesses:**

**Strengths**

- The hierarchical design of the instruction dataset is well thought out and supports improved reasoning capabilities.
- The integration of 2D and 3D molecular representations via a blending module effectively captures both topological and spatial features of molecules. This design choice is supported by ablation studies.

**Weaknesses**

- The paper primarily reuses existing tools such as GPT-4o for data generation and filtering, pre-trained molecular encoders, and Q-Former. The main novelty lies more in the engineering effort and the construction of the dataset rather than in introducing new models or learning algorithms.
- The paper uses GPT-4o both to generate molecular reasoning data and to evaluate the factual accuracy of the generated samples. While this is a practical approach given GPT-4o’s strong domain knowledge, I am concerned about potential risks of hallucination. If GPT-4o produces factually incorrect or misleading descriptions during generation, it may not always be able to reliably identify these errors during the evaluation phase, since both tasks rely on the same model’s internal knowledge. I suggest the authors discuss this limitation explicitly and, if possible, consider additional validation strategies, such as incorporating human expert review.
- The instruction dataset is a central contribution of the paper. However, the details of how this dataset is constructed are largely placed in the appendix. I recommend moving the dataset construction process into the main text and expanding the discussion to make the contribution clearer and more accessible to readers.

- *Minors:* The paper refers to the "string version of the molecule" and clarifies that this corresponds to the IUPAC name. Since IUPAC names are less commonly used as string representations in computational chemistry compared to formats like SMILES, SELFIES, or InChI, it would be clearer and more precise for the authors to explicitly state "IUPAC name" instead of the more ambiguous "string version." This would help avoid confusion for readers familiar with standard molecular string formats.

---

> ### Author Rebuttal · Authors · 2025-07-30
>
> We sincerely thank you for your constructive and helpful comments. We appreciate the positive comments.
> - The hierarchical design of the dataset is well thought out and supports improved reasoning capabilities.
> - The integration of 2D and 3D molecular representations via a blending module effectively captures both topological and spatial features.
>
> We initially address your concerns below.
>
> ---
> *Comment 1*: The paper primarily reuses existing tools such as GPT-4o for data generation and filtering, pre-trained molecular encoders, and Q-Former. The main novelty lies more in the engineering effort and the construction of the dataset rather than in introducing new models or learning algorithms.
>
> *Response 1*: We propose a novel framework to build molecular LLMs, making notable contributions in fourfold: 1) **novel dataset construction pipeline for molecules**, 2) **model architecture**, 3) **resulting foundational model, Mol-LLaMA**, and 4) **further utilization**.
>
> > ### Dataset Construction Pipeline
> - Motivation-level: **We first tackle that existing databses and dataset construction pipelines are not appropriate to build the molecular LLMs**, identifying the following two challenges: 1) molecular LLMs should learn a wide-ranging knowledge encompassing chemical and biological features as this knowledge is domain-specific and 2) unlike other modalities, molecular features are not directly apparent from structures, requiring complex reasoning and interpretability [lines 31-52, 112-125, and Table 2].
> - Methodology-level: To solve these limitations, **we design novel data types that have not been explored in the literature**, based on our observation of the hierarchical relationships between molecular structures and their features.
>
> > ### Model Architecture
> - Motivation-level: **We tackle that the current molecular LLMs rely on a single type of molecular representations**, often failing to correctly predict the molecular properties [lines 52-54].
> - Methodology-level: We observe that 2D and 3D molecular encoders have their distinct advantages. Thus, **we opt to utilize both molecular representations [lines 167-172] and propose an effective model architecture, the blending module, that can integrate their advantages [lines 173-178], which have not been covered by previous works.**
>
> > ### Resulting Foundational Model: Mol-LLaMA
>
> Our resulting model, Mol-LLaMA, exhibits several key strengths as follows:
> - General understanding of molecular features: **Mol-LLaMA accurately predicts the general molecular features compared to other molecular LLMs** as shown in Table 3 and 4, showcasing its potential as a general-purpose assistant for molecular analysis.
> - Explainability and Reasoning Capabilities: Unlike other molecular LLMs, **Mol-LLaMA provides interpretable responses with informative reasoning processes**  as shown in Table 1, 9, 10 11, and 17, enhancing the reliability and thus further opening its utilization for practical fields.
>
> > ### Further Utilization
>
> We would like to emphasize that **our dataset construction pipeline and the model architecture can be further utilized to build other LLMs specialized in understanding the biomolecules such as proteins, RNAs and their complexes, which are notable contributions for the AI4Science field** [lines 306-309]. Please note that we will publicly release all materials, including code, data, and model weights, as a state-of-the-art foundational model for molecular fields.
>
> ---
> *Comment 2*: While this is a practical approach given GPT-4o’s strong domain knowledge, I am concerned about potential risks of hallucination.
>
> *Response 2*: We appreciate your valuable comment. First, we would like to clarify that **our instruction dataset generation is grounded in the annotated molecular features and IUPAC names provided by PuChem, which helps mitigate the risk of hallucination**.
>
> Additionally, **we adopt GPT-4o due to its demonstrated capabilities in scientific domains, as evidenced by prior studies [1], which support its reliability in this context**. In contrast, other LLMs have not been rigorously analyzed in the molecular science domain, making them less reliable for generating and evaluating data.
>
> To show the effect of other LLMs, we filtered data using Qwen1.5-14B, Gemma-1.5-12B-IT, Llama-3.1-8B-Instruct and GPT-4o, selecting samples rated 4 by all. As shown in Table R1, this quadruple-filtered version (Mol-LLaMA$^*$) yielded no significant gains, suggesting these LLMs may wrongly discard correct data and limit model scope. We will include the results in the final revision.
>
> Although current LLMs are still limited, we would like to emphsize that **our dataset construction pipeline offers a notable contribution for building molecular LLMs by being readily adapted to future LLMs with more extensive knowledge in scientific domains**.
>
>
> |Models|Help.|Relev.|Acc.|Details|Overall|Help.|Relev.|Acc.|Details|Overall|Help.|Relev.|Acc.|Details|Overall|
> |-|-|-|-|-|-|-|-|-|-|-|-|-|-|-|-|
> |Mol-LLaMA|1.126|1.145|1.154|1.090|1.125|1.224|1.266|1.302|1.211|1.251|1.578|1.840|2.030|1.528|1.744|
> |Mol-LLaMA$^*$|1.004|1.024|1.035|0.941|1.006|1.143|1.193|1.255|1.113|1.179|1.605|1.858|2.097|1.569|1.801|
>
> *Table R1: Quantitative evaluation on structural (left 5 criteria), chemical (middle 5 criteria), and biological (right 5 criteria) understanding.*
>
> [1] AI4Science, Microsoft Research, and Microsoft Azure Quantum. "The impact of large language models on scientific discovery: a preliminary study using gpt-4." arXiv preprint arXiv:2311.07361 (2023).
>
> ---
> *Comment 3*: If possible, consider additional validation strategies, such as incorporating human expert review.
>
> *Response 3*: We thank you for your valuable suggestion. However, **gathering a sufficient number of qualified experts for accurately evaluating data quality is highly challenging**. First, assessing the generated molecular descriptions requires interdisciplinary expertise across chemistry, biochemistry, biology, and pharmacology. Additionally, some properties may not be annotated in public databases like PubChem, meaning factually correct descriptions could appear unsupported. The inherent ambiguity in certain molecular properties or descriptions may also introduce evaluator bias, as experts from different backgrounds might interpret the same description differently.
>
> ---
> *Comment 4*: I recommend moving the dataset construction process into the main text and expanding the discussion to make the contribution clearer and more accessible to readers.
>
> *Response 4*: Due to the page limit, we explained the core components of our dataset construction pipelines, including context types, data types, and filtering process, postponing the details such as prompts to the appendix. We will move detailed explanations of the dataset construction pipeline to the main paper in the final revision.
>
> ---
> *Comment 5*: It would be clearer and more precise for the authors to explicitly state "IUPAC name" instead of the more ambiguous "string version."
>
> Response 5. We will revise the term “string representation” to its specific type, such as SMILES or IUPAC name for clarity.
>
> ---
> *Comment 6*: How well justified is the choice of molecular encoders (MoleculeSTM for 2D, UniMol for 3D)? Were alternative encoders considered or compared?
>
> *Response 6*: We adopt MoleculeSTM as it inherently captures molecular semantics through contrastive learning between 2D molecular structures and textual descriptions. Prior studies [2] have shown that contrastive learning with textual descriptions shows better performance than reconstruction-based training for building multimodal LLMs.
>
> UniMol is a state-of-the-art molecular encoder across both 2D and 3D representations. Although it is trained with a reconstruction loss, its use of transformer architectures enables strong representational expressiveness.
>
> [2] McKinzie, Brandon, et al. "Mm1: methods, analysis and insights from multimodal llm pre-training." European Conference on Computer Vision. Cham: Springer Nature Switzerland, 2024.
>
> ---
> We thank the reviewer for your time and feedback and hope that our responses have addressed your concerns. If you have any further questions, we are happy to address them. If your concerns are appropriately addressed, we hope you kindly consider updating the rating accordingly.

---

> ### Comment · Reviewer_voqH · 2025-08-05
>
> 1. Thank you for your detailed response. I acknowledge and appreciate the contribution of your dataset, which I had already mentioned in my initial comment as a key contribution of the paper.
>
> * The model architecture mostly combines existing 2D and 3D encoders (MoleculeSTM and UniMol) rather than introducing a fundamentally new design. This kind of integration, while practical, isn’t a strong novelty on its own. I’m not saying it’s bad to use others’ encoders, but I want to make it clear that there’s no novelty in this point. The Blending Module contains self-attention and cross-attention layers, and the Q-Former are also not new.
> * Regarding the foundational model as a novelty, Mol-LLaMA’s strengths in molecular understanding and reasoning seem to come mainly from the training data (data-driven) rather than from new modeling or algorithms. Introducing a new model that performs well is common but doesn’t automatically imply a method advance.
> * The further utilization you mention also depends on these data-driven capabilities. Overall, the main contribution appears to be the dataset and its construction pipeline, not the modeling methods.
>
> 2. Thank you for sharing insights on the performance of other LLMs, that's very helpful. From my own experience, GPT4o’s ability to generate molecular descriptions is hallucination-prone and only performs reliably only on common molecules. For instance, a single molecule can be represented by multiple SMILES strings, and depending on which SMILES is used as input, the model’s output can vary largely. Additionally, GPT4o may mistakenly consider two distinct molecules similar due to the one-to-many mapping between molecules and their SMILES representations. This is my prompt for ChatGPT (GPT4o-based):
>
> > Given 2 molecules:
> CCCCOC(=O)CCC(C(=O)OCCCC)NC(=O)C1=CC=C(C=C1)N(C)CC2=CN=C3C(=N2)C(=NC(=N3)N)N
> and
> CCCCOC(=O)CCC(NC(=O)c1ccc(N(C)Cc2cnc3(N)nc(N)c3n2)cc1)C(=O)OCCCC.
> Are they represent the same molecule?
> Only answer Yes or No.
>
> The model answers “Yes” — even though the second molecule is not a valid structure.
> > Given 2 molecules:
> C(C1C(C(C(C(=O)O1)O)O)O)OP(=O)(O)O
> and
> O=C1OC(COP(=O)(O)OCCC)C(O)C(O)C1O.
> Are they represent the same molecule?
> Only answer Yes or No.
>
> The model answers Yes even when they are different molecules.
>
> Regarding the 2D and 3D encoders you employed, both are transformer-based architectures rather than graph neural networks (GNNs). This means they still heavily rely on sequentialized molecular representations, which inherently can vary for the same molecule. That's why I had a question about choosing molecular encoders. One GNN encoder may alleviate this problem because molecule to molecular graph is one-one mapping. I also noticed there was no mention of training on canonical SMILES, which may contribute to the unreliability of GPT4o’s generated results.
>
> Given that your model is trained multi-level descriptions (structure, function), could you test whether queries using two different SMILES strings for the same molecule produce consistent results for a given function (e.g., BBBP prediction)? This would be particularly important since you are not using human evaluation.
>
> 3. Regarding my comment on additional validation strategies, what I am suggesting is human validation of the LLM outputs on a random subset of your dataset — not manual annotation of the entire dataset. You argue that factually correct (generated) descriptions could appear unsupported is much less likely than the model generating incorrect descriptions. That’s why I recommend human evaluation: to better understand how reliable the generated descriptions actually are. While it can be challenging, it’s certainly not impossible. I believe experts can perform this validation to some extent, and doing so would only strengthen your dataset. I understand that this validation might not be feasible at this stage — it’s just a suggestion and doesn’t affect my overall assessment of your paper.

---

> ### Author Response · Authors · 2025-08-05
> **Thank you for engaging in the discussion**
>
> We sincerely appreciate the reviewer’s valuable comments and engaging in the discussion. We further address your concerns below.
>
> ---
> *Comment 7*: Regarding the novelty of the model architecture.
>
> *Response 7*: We thank the reviewer for the insightful feedback. While our model is built upon the existing components, **our originality lies in the straightforward yet effective model architecture that effectively leverages and integrates multiple molecular representations to address key limitations in molecular LLMs**. In this sense, we respectfully believe that our approach offers a novel and well-reasoned combination that demonstrates practical relevance and empirical gains, in line with the broader perspective on originality.
>
> ---
> *Comment 8*: Mol-LLaMA’s strengths in molecular understanding and reasoning seem to come mainly from the training data (data-driven) rather than from new modeling or algorithms.
>
> *Response 8*: To the best of our knowledge, **the significance of the instruction dataset is becoming crucial**, as recent work underscores the central role of instruction datasets in endowing LLMs with new skills [3]. In this spirit, **we present a novel algorithmic pipeline for constructing instruction datasets**, especially suited for scientific problems that require domain-specific knowledge and complex reasoning capabilities. Additionally, our experimental comparisons and ablation study provide valuable insights into the effects of the role of instruction datasets and the types of knowledge necessary for developing advanced LLMs tailored to scientific problem solving.
>
> [3] Yue, Yang, et al. "Does reinforcement learning really incentivize reasoning capacity in llms beyond the base model?." arXiv preprint arXiv:2504.13837 (2025).
>
> ---
> *Comment 9*: Regarding the foundational model as a novelty.
>
> *Response 9*: We respectfully believe that **proposing a foundational model tailored for molecular reasoning is a significant contribution to advancing the AI4Science field**. Foundational models have the potential to generalize across multiple molecular tasks, enable transfer learning for downstream scientific applications, and further induce future works, such as training-free molecular reasoning enhancement or building an end-to-end framework that combines LLM capabilities with external scientific resources. We believe these opportunities further highlight the importance of our proposed foundational model.
>
> ---
> *Comment 10*: GPT-4o’s hallucinations on SMILES when generating data.
>
> *Response 10*: We would like to clarify that **we use the annotated IUPAC names to generate the data to alleviate the hallucination problems**. The design choice is motivated by the analyses that IUPAC names are the most interpretable string representations, as studied in the recent work [4]. Additionally, at the time of our study, GPT-4o was the most suitable model for automating the dataset construction pipeline, as it is the only model whose capabilities in scientific problem solving have been extensively analyzed. **While GPT-4o is not without limitations, our dataset construction pipeline leverages its strengths to incorporate comprehensive molecular knowledge and reasoning capabilities, as you mentioned**.
>
>
> [4] Microsoft Research AI4Science, and Microsoft Azure Quantum. "The impact of large language models on scientific discovery: a preliminary study using gpt-4." arXiv preprint arXiv:2311.07361 (2023).
>
> ---
> *Comment 11*: Sequentialized molecular representations of encoders might introduce error.
>
> *Response 11*: **Our model architecture ensures permutation invariance by avoiding reliance on sequential molecular representations.** Specifically, MoleculeSTM is built on a GNN architecture (i.e., GIN) and UniMol utilizes interatomic distances as features without using positional embeddings from the original transformer architecture, thereby maintaining permutation equivariance. Additionally, Q-former ensures permutation invariance through its cross-attention mechanisms, as discussed in lines 182-183. Please note that we use the molecular structures rather than SMILES representations as inputs to mitigate the issue that a single molecular structure can have multiple SMILES forms.
>
>
> ---
> *Comment 12*: Human evaluation on a subset of our instruction dataset.
>
> *Response 12*: Thank you for your valuable suggestion. We are currently conducting the human evaluation and will share the results as soon as they are finalized.
>
> ---
> We sincerely thank you again for your time and effort in reviewing our paper. If you have further questions, we are happy to address them.

---

> > ### Author Response · Authors · 2025-08-07
> > **We share the finalized human evaluation results**
> >
> > Dear Reviewer voqH,
> >
> > As the human evaluation is finalized, we would like to politely draw your attention again.
> >
> > ---
> > *Comment 12*: Human evaluation on a subset of our instruction dataset.
> >
> > *Response 12*: We conducted a human evaluation on fifteen samples of our detailed structural descriptions, involving nine experts specializing in chemistry and biology. The experts were asked to rate the accuracy of each structural description on a 4-point scale. In Table R2, we report the Pearson and Spearman correlation coefficients between the average human evaluation scores and the scores given by GPT-4o. Both coefficients are close to 1 and associated with low $p$-values, **indicating a strong and statistically significant agreement between human evaluations and GPT-4o's assessments. Notably, the average human score for the subset of data that received a score of 4 from GPT-4o is 3.71, further demonstrating the high reliability of our instruction dataset.**
> >
> > |Pearson|$p$-value|Spearman|$p$-value|
> > |-|-|-|-|
> > |$0.96$|$2.68\times10^{-8}$|$0.91$|$3.43\times10^{-6}$|
> >
> > *Table R2: Pearson correlation coefficient between the scores of human and GPT-4o.*
> >
> > ---
> > We sincerely appreciate your thoughtful feedback and hope that the human evaluation results have addressed your concerns. If you have further feedback and questions, we would be happy to respond. If your concerns are appropriately resolved, we kindly ask you to consider updating your ratings.

---

> > > ### Comment · Reviewer_voqH · 2025-08-08
> > >
> > > Thank you for your response and for addressing my concerns, especially regarding the human evaluation.
> > >
> > > While I still believe the model-side method lacks novelty, I acknowledge that the newly introduced dataset is valuable. The inclusion of preliminary human evaluation results with a fairly high R2 is promising. Given this, I am willing to raise my rating, as I believe this work could make a useful contribution to the research community.

---

### Official Review · Reviewer_6V81 · 2025-06-26

**Clarity:** 4
**Significance:** 2
**Originality:** 3
**Rating:** 4
**Confidence:** 3

**Summary:**

In this paper, Mol-LLaMA, a large molecular language model that grasps the general knowledge centered on molecules and exhibits explainability and reasoning ability, is introduced. To develop the model, a novel, large-scale instruction dataset (Mol-LLaMA-Instruct) is constructed with GPT-4o, featuring three specific data types designed to teach foundational molecular knowledge and causal relationships. To improve the performance, the authors design a "blending module" that integrates information from both 2D and 3D molecular encoders to create a more comprehensive structural representation.  The paper demonstrates through extensive experiments that Mol-LLaMA outperforms existing models, including GPT-4o, in tasks such as molecular understanding and property prediction.

**Questions:**

1. How effective are the molecular representations learned by the proposed model for direct application in downstream tasks? Furthermore, what are its specific advantages over previous unified representation models, such as in [1]?
2. Regarding the evaluation methodology using GPT-4o as a judge: The paper describes a pairwise comparison to derive a relative score. Was one of the two assistants consistently the GPT-4o model itself, acting as a reference baseline? The prompt instructs the judge to avoid order effects, but were procedural safeguards, such as randomizing the response order, implemented to mitigate positional bias? Finally, considering these factors, how robust is this relative scoring method compared to an absolute grading system where each response would be evaluated individually against a fixed rubric?
3. Prompt Specificity and Bias: The model was trained and tested on a specific set of instruction formats. To what extent does this in-distribution testing create an advantage for Mol-LLaMA over other models that were not trained on these specific prompts?
4. Zero-Shot vs. Fine-Tuning: On the MoleculeQA benchmark, all models were fine-tuned. Would a zero-shot evaluation on this task be possible, and what might it reveal about the model's intrinsic generalization capabilities without task-specific training?
5. Property Prediction Benchmarking: For the molecular property prediction tasks, the comparison could be expanded since Mol-LLaMA is finetuned on the dataset. Have you considered: Including more datasets from the same class (e.g., Tox21, HIV)? Benchmarking against smaller, state-of-the-art models that are specialized for these tasks such as [2] ?
6. Using the Area Under the ROC Curve (AUROC) metric in addition to accuracy, as it can be more informative for classification tasks?
7. Evaluation of Baselines: In the property prediction results (Table 4), the "Fidelity" and "Helpfulness" scores are not reported for the GPT-4o baseline. Could you clarify why these metrics were omitted for the baseline model?
8. Does the model support inputs involving multiple molecules at once? The general LLMs such as GPT-4o can process SMILES of multiple molecules.
9. Influence of Molecular Conformation: Given the use of a 3D encoder, how sensitive is the model's performance to different input molecular conformations? How effectively does it handle molecules with high conformational flexibility?
10. How well does the model follow instructions and generalize to tasks beyond those it was explicitly trained on?
11. Practical Utility: As a general-purpose tool, what are some concrete examples of how Mol-LLaMA could assist chemists or biologists in their day-to-day research workflows?
12. Comparison with Other Models: The current benchmarks include models of a similar scale. How would Mol-LLaMA be expected to perform against significantly larger foundation models such as GPT-4.5? Could a comparison be made against other existing general-purpose (multi-modal) chemical language models that were not included in this study, such as [3, 4]?
13. Open-Source Plans: Are there plans to release the model weights, training code, and the Mol-LLaMA-Instruct dataset to the public to support reproducibility and further community research? I don’t find the links in this paper.

I recognize that my previous questions cover a wide range of topics, and it may not be feasible to address every point exhaustively. Therefore, you can focus your response on clarifying and defending the primary contribution of your work. The key is to demonstrate the unique value proposition of Mol-LLaMA. For instance:
If the core contribution is superior molecular representation, please provide more targeted evidence to demonstrate this superiority over existing methods.
If the central claim is that Mol-LLaMA is a general-purpose foundational model for chemistry, its evaluation should include a broader set of tasks to prove its versatility and superiority over other general-purpose chemical models.
If, however, Mol-LLaMA excels as a specialized model for a particular task, the most convincing comparison would be against state-of-the-art (SOTA) models dedicated to that task.
Clearly articulating and substantiating the unique value would strengthen the paper and my assessment of it.

[1]Zhu, Jinhua, et al. "Unified 2d and 3d pre-training of molecular representations." Proceedings of the 28th ACM SIGKDD conference on knowledge discovery and data mining. 2022.
[2]Zhang, Zaixi, et al. "Motif-based graph self-supervised learning for molecular property prediction." Advances in Neural Information Processing Systems 34 (2021): 15870-15882.
[3]Zhang, Di, et al. "Chemllm: A chemical large language model." arXiv preprint arXiv:2402.06852 (2024).
[4]Li, Junxian, et al. "Chemvlm: Exploring the power of multimodal large language models in chemistry area." Proceedings of the AAAI Conference on Artificial Intelligence. Vol. 39. No. 1. 2025.

**Ethical Concerns:**

["NO or VERY MINOR ethics concerns only"]

**Final Justification:**

I still hold the reservation that the proposed method may not be the most effective path toward creating a truly general-purpose foundational model for chemistry. In my view, more convincing routes toward that goal would likely involve continued pre-training on large-scale chemical and biological corpora or learning from real-world feedback. For this reason, I believe the long-term significance of this specific approach is somewhat limited.

Nevertheless, considering the strong performance demonstrated across multiple tasks in the manuscript, I raise my rating to 4.

**Limitations:**

1. No Generation abilities: Mol-LLaMA is designed primarily for molecular analysis and understanding.  It currently lacks the capability to perform generative tasks, such as designing novel molecules with desired properties. The authors have to consider how this model can help the improvement of chemistry or biology,
2. Dataset Filtering: While the instruction dataset was filtered for factual accuracy using GPT-4o as a judge, this process is not infallible.  There is still a possibility that incorrect or subtly flawed information remains in the samples used for training.

**Quality:**

2

**Strengths And Weaknesses:**

strength
1. Comprehensive Instruction Dataset: A core strength is the systematic design of the Mol-LLaMA-Instruct dataset. Instead of relying on existing task-specific datasets, the authors created three new data types: "Detailed Structural Description," "Structure-to-Feature Relationship Explanation," and "Comprehensive Conversation."  This approach explicitly aims to build a foundational understanding of molecular principles and enhance the model's reasoning capabilities.
2. Innovative Model Architecture: The proposed 2D-3D blending module is an architectural contribution.  Recognizing that 2D and 3D representations offer complementary information (e.g., bond connectivity vs. spatial arrangement), the model uses a cross-attention mechanism to integrate them, which is shown to reduce errors and improve the understanding of molecular structures compared to using a single encoder or simple concatenation.

weakness
1. Reliance on GPT-4o for Data and Evaluation: The instruction dataset was generated using GPT-4o, and the quality of the generated responses was also evaluated using GPT-4o as a judge.  This heavy reliance on a proprietary, closed-source model introduces potential issues:
Inherited Biases: Any factual errors, biases, or limitations present in GPT-4o could be propagated into the training data and, consequently, into Mol-LLaMA.
Evaluation Circularity: Using the same family of models for both data generation and evaluation can lead to results that favor the style and knowledge base of that model, potentially inflating performance scores.
2. Lack of Statistical Significance Testing: The authors state that due to high computational costs, they did not train models multiple times.  While understandable, this makes it harder to assess the robustness and generalizability of the performance gains. At least, the authors should use temperature sampling and test the results multiple times.

---

> ### Author Rebuttal · Authors · 2025-07-30
>
> We sincerely thank you for your constructive and helpful comments. We appreciate the positive comments.
> - The design of the three new data types and the resulting instruction dataset builds a foundational understanding of molecular principles and enhances the reasoning capabilities
> - The proposed 2D-3D blending module is innovative, reduces errors, and improves the understanding of molecular structures.
>
> Due to the character limit, we initially address your major concerns and some of your questions below.
>
> ---
> *Comment 1*: The closed-source model (i.e., GPT-4o) could introduce potential issues: Factual errors or biases could be propagated into the training data.
>
> *Response 1*: We appreciate your valuable comment. **While we aimed to select reliable LLMs to ensure data quality, to the best of our knowledge, alternative models lack extensive analysis capabilities for molecular and scientific reasoning**. In other words, GPT-4o [1] remains the only model with demonstrated reliability in these domains, making it a more suitable choice despite its closed-source nature.
>
> Further, we investigate the effect of other LLMs for evaluating the data quality by filtering with four LLMs: GPT-4o, Qwen1.5-14B, Gemma-1.5-12B-IT, and Llama-3.1-8B-Instruct, selecting samples rated 4 by all. As shown in Table R1, this quadruple-filtered version (Mol-LLaMA$^*$) yielded no significant gains, suggesting these LLMs may wrongly discard correct data and limit knowledge scope. We will include the results in the final revision.
>
> Despite the limitations of current LLMs, we would like to emphasize that **our dataset construction pipeline is a notable contribution for building molecular LLMs as it can be readily applied with upcoming LLMs exhibiting more extensive knowledge for scientific domains**.
>
> |Models|Help|Relev|Acc|Details|Overall|Help|Relev|Acc|Details|Overall|Help|Relev|Acc|Details|Overall|
> |-|-|-|-|-|-|-|-|-|-|-|-|-|-|-|-|
> |Mol-LLaMA|1.126|1.145|1.154|1.090|1.125|1.224|1.266|1.302|1.211|1.251|1.578|1.840|2.030|1.528|1.744|
> |Mol-LLaMA$^*$|1.004|1.024|1.035|0.941|1.006|1.143|1.193|1.255|1.113|1.179|1.605|1.858|2.097|1.569|1.801|
>
> *Table R1: Quantitative evaluation on structural (left 5 criteria), chemical (middle 5 criteria), and biological (right 5 criteria) understanding.*
>
> [1] Microsoft Research AI4Science and Microsoft Azure Quantum. "The impact of large language models on scientific discovery: a preliminary study using gpt-4." arXiv:2311.07361 (2023).
>
> ---
> *Comment 2*: Using the same family of models for both data generation and evaluation can lead to results that favor their style and knowledge base, potentially inflating performance scores.
>
> *Response 2*: We thank you for your valuable comment. To investigate whether the same family favors the style and knowledge base, we conduct the evaluation using four LLMs: GPT-4o, Qwen3-14B, Gemma-3-12B-IT, and Llama-3.1-8B-Instruct. As shown in Table R2, **Mol-LLaMA still outperforms all baselines, including GPT-4o, with consistent trends observed in Table 3, implying that GPT-4o’s evaluation is not biased rather agrees with other LLMs**. We will include these results in the final revision.
>
> |Models|Help|Relev|Acc|Details|Overall|Help|Relev|Acc|Details|Overall|Help|Relev|Acc|Details|Overall|
> |-|-|-|-|-|-|-|-|-|-|-|-|-|-|-|-|
> |**Llama2-based**|
> ||
> |Llama-2-7B-Chat|0.372|0.374|0.264|0.359|0.328|0.534|0.520|0.405|0.551|0.492|0.479|0.408|0.345|0.598|0.441|
> |Mol-Instructions|0.217|0.239|0.252|0.131|0.202|0.242|0.274|0.301|0.145|0.229|0.319|0.376|0.416|0.201|0.315|
> |LlasMol|0.278|0.306|0.304|0.204|0.265|0.295|0.335|0.312|0.215|0.280|0.332|0.378|0.443|0.260|0.341|
> |3D-MoLM|0.674|0.658|0.564|0.719|0.643|0.784|0.788|0.701|0.798|0.762|0.898|0.945|0.953|0.911|0.923|
> |LLaMo|0.306|0.407|0.443|0.179|0.312|0.345|0.464|0.524|0.197|0.356|0.454|0.630|0.802|0.236|0.490|
> |Mol-LLaMA|1.042|1.063|1.063|0.960|1.038|1.124|1.179|1.212|1.044|1.146|1.376|1.607|1.683|1.167|1.468|
> ||
> |**Llama3.1-based**|
> ||
> |Llama3.1-8B|0.669|0.680|0.560|0.648|0.634|0.698|0.693|0.600|0.695|0.667|0.705|0.663|0.617|0.732|0.674|
> |Mol-Instructions|0.261|0.334|0.369|0.152|0.263|0.265|0.350|0.396|0.149|0.269|0.358|0.468|0.577|0.191|0.367|
> |3D-MoLM|0.835|0.874|0.794|0.782|0.817|0.902|0.973|0.902|0.829|0.900|1.114|1.295|1.346|0.970|1.180|
> |LLaMo|0.480|0.616|0.560|0.301|0.473|0.401|0.534|0.560|0.245|0.414|0.596|0.788|0.834|0.311|0.603|
> |Mol-LLaMA|1.052|1.081|1.092|0.964|1.049|1.139|1.192|1.225|1.068|1.163|1.459|1.699|1.801|1.210|1.555|
> |Mol-LLaMA$^*$|1.001|1.026|1.030|0.911|0.997|1.113|1.170|1.205|1.034|1.139|1.421|1.640|1.741|1.223|1.522|
>
> *Table R2:  Quantitative evaluation on structural (left 5 criteria), chemical (middle 5 criteria), and biological (right 5 criteria) understanding using diverse evaluators.*
>
> ---
> *Comment 3*: Lack of statistical significance testing makes it harder to assess the robustness and generalizability.
>
> *Response 3*: We appreciate your valuable suggestion. However, **we adopt greedy decoding, as it aligns with real-world applications where users prioritize the most confident and reliable responses from LLMs, specifically for the scientific problems**. Instaed, we perform the LLM evaluation three times to assess the quality of the generated responses.
>
> Furthermore, we apply the temperature sampling with three runs and report the averaged metrics in Table R3 using the same four evaluator LLMs in *Response 2*. **Performance is maintained and Mol-LLaMA consistently outperforms the baselines, including GPT-4o, indicating that Mol-LLaMA robustly and generally shows superior performance**.
>
> |Models|Help|Relev|Acc|Details|Overall|Help|Relev|Acc|Details|Overall|Help|Relev|Acc|Details|Overall|
> |-|-|-|-|-|-|-|-|-|-|-|-|-|-|-|-|
> |Llama-3.1-8B-Instruct|0.650|0.671|0.555|0.631|0.623|0.699|0.691|0.619|0.719|0.675|0.724|0.704|0.657|0.761|0.706|
> |Mol-Instructions|0.258|0.327|0.355|0.149|0.258|0.275|0.345|0.391|0.153|0.275|0.355|0.456|0.583|0.190|0.370|
> |3D-MoLM|0.694|0.679|0.575|0.734|0.663|0.756|0.761|0.665|0.771|0.731|0.869|0.886|0.852|0.874|0.870|
> |LLaMo|0.475|0.592|0.543|0.306|0.465|0.405|0.526|0.554|0.241|0.413|0.595|0.746|0.795|0.311|0.588|
> |Mol-LLaMA|1.035|1.046|1.068|0.993|1.040|1.118|1.164|1.202|1.066|1.140|1.390|1.588|1.693|1.206|1.483|
>
> *Table R3: Quantitative evaluation on structural (left 5 criteria), chemical (middle 5 criteria), and biological (right 5 criteria) understanding with temperature sampling*
>
> ---
> *Comment 4*: As a general-purpose tool, what are some concrete examples of how Mol-LLaMA could assist chemists or biologists in their research workflows?
>
> *Response 4*: **We carefully designed the experiments to show the effectiveness of Mol-LLaMA in scientific research workflows**, focusing on three key points: **1) Molecular analysis**, **2) Interactive manipulation**, and **3) Task specialization**.
>
> > ### Analysis of De novo Molecules
>
> Chemists, biologists, and drug developers frequently encounter novel molecules that lack annotations in public databases. In such cases, Mol-LLaMA enables initial assessments by providing insights into physicochemical properties and likely biological functions. Tables 3 and 4 demonstrate Mol-LLaMA’s effectiveness in this utilization, showcasing its superior ability in accurately predicting molecular features with helpful explanations.
>
> > ### Interactive Manipulation
>
> Another key strength of Mol-LLaMA is its interactive capability, allowing users to guide its outputs through prompts. For the PAMPA task, we devise “with task info” prompting that injects the task-specific information reflecting real-world scenarios where users guide LLM outputs. As shown in Table 4, Mol-LLaMA built upon LLaMA3.1, shows performance gains when prompted with task-specific information, demonstrating its ability to effectively handle diverse user prompts.
>
> > ### Task Specialization
>
> Mol-LLaMA can be further fine-tuned for specific tasks, making it well-suited for researchers and developers working in highly specialized domains such as drug discovery. We validate Mol-LLaMA’s effectiveness in task transfer settings of Tables 5, 12, and 13, demonstrating superior performance compared to baseline models.
>
> ---
> *Comment 5*: To what extent does this in-distribution testing create an advantage for Mol-LLaMA over other models that were not trained on these specific prompts?
>
> *Response 5*: **In the PAMPA task, we assess Mol-LLaMA’s generalization ability using prompts that were entirely different from those seen during training**, as shown in Table 32. These settings show that Mol-LLaMA is capable of reasoning and generating accurate answers beyond the training prompts, showcasing its generalization ability.
>
> ---
> *Comment 6*: Have you considered more datasets such as MoleculeNet?
>
> *Response 6*: We opted for the PAMPA task as it is well-suited to evaluate whether molecular LLMs truly understand molecular structures and features. In contrast, tasks in MoleculeNet require external knowledge beyond the molecule itself such as the structure of the blood-brain barrier for the BBBP task or nuclear receptors for the Tox21 task which heavily influenced by the internal knowledge of the base LLM, making fair comparisons difficult.
>
> Nevertheless, we include a comparison on the BBBP task from MoleculeNet in Table 8, where Mol-LLaMA outperforms existing molecular LLMs built upon the same model architectures.
>
> ---
> *Comment 7*: Are there plans to release the model weights, training code, and the Mol-LLaMA-Instruct dataset?
>
> *Response 7*: We will release the Mol-LLaMA weights, training code, and Mol-LLaMA-Instruct dataset to facilitate further research in molecular foundation models and practical usage for the scientific domain.
>
> ---
> We thank the reviewer for your time and feedback and hope that our responses have addressed your concerns. If you have any further questions, we are happy to address them. If your concerns are appropriately addressed, we hope you kindly consider updating the rating accordingly.

---

> > ### Comment · Area_Chair_QMDC · 2025-08-05
> >
> > Dear Reviewer 6V81,
> >
> > Would you mind to have a check over authors' replies?
> >
> > AC

---

> > ### Comment · Reviewer_6V81 · 2025-08-06
> >
> > Thank you for your detailed response. Based on your clarifications, I now understand that the positioning of this work is as a general-purpose foundational model for chemistry. Your answers are compelling, and the additional experiments you provided have successfully addressed my previous concerns on several points.
> >
> > Could you also provide responses to a few other questions related to this "general-purpose foundational model" concept, specifically Questions 2, 8, 9, and 12.
> >
> > If these points can be satisfactorily addressed, I would be happy to raise my rating for the manuscript.

---

> ### Author Response · Authors · 2025-08-07
> **Thank you for engaging in the discussion (Part 1)**
>
> We sincerely appreciate your thoughtful response and engagement in the discussion. We are grateful for the opportunity to address the remaining questions and further deepen the understanding of our work.
>
> We address your remaining questions below.
>
> ---
> *Comment 8*: Regarding the evaluation setting: 1) were GPT-4o consistently used as a reference baseline? and 2) Were procedural safeguards implemented?
>
> *Response 8*: Yes, we leverage GPT-4o as a consistent reference baseline. And we randomly shuffled the response order for each evaluation and reported the average of three evaluations to mitigate the positional bias.
>
> ---
> *Comment 9*: How robust is this relative scoring method compared to an absolute grading system?
>
> *Response 9*: While it is difficult to determine which scoring method is generally more robust, **we chose the relative scoring method since it best suits our experimental objectives**. Specifically, as studied in recent works [2,3], the relative scoring method is effective for comparing and identifying better responses, especially in the cases of qualitative assessments. As our experiments include the quality evaluation such as helpfulness, relevance, level of detail, and fidelity, we adopted the relative scoring method rather than the absolute grading system.
>
> ---
> *Comment 10*: Does the model support inputs involving multiple molecules at once? The general LLMs such as GPT-4o can process SMILES of multiple molecules.
>
> *Response 10*: Our model can process and understand multiple molecules thanks to the nature of LLMs that understand diverse contexts. Due to the lack of an appropriate benchmark to assess this capability, we experimentally address the challenge by employing a well-established prompting method: In-context learning with examples.
>
> **Experimental Setup**
> - For the PAMPA task, we randomly selected $k$ pairs of molecules and their corresponding labels from the training dataset.
> - During inference, we used the same input-label pair examples in the same order across models to minimize other biases such as positional, textual, and molecular biases. This setup allows us to focus solely on evaluating each model’s ability to understand multiple molecules.
>
> **Results and Analysis**
>
> As shown in Table R4, Mol-LLaMA attains a consistent performance improvement as the number of in-context examples increases, **demonstrating its capability to effectively process and understand multiple molecules and their corresponding labels**. In contrast, GPT-4o does not effectively utilize multiple molecules, showing the performance declines with one or three in-context examples. We will include these results in the final revision.
>
> |Models|0-shot|1-shot|3-shot|5-shot|
> |-|-|-|-|-|
> |Mol-LLaMA|63.55|73.46|77.64|80.10|
> |GPT-4o|48.65|31.45|46.68|57.74|
>
> *Table R4: Accuracy on the PAMPA task using different numbers of in-context learning examples.*
>
> ---
> *Comment 11*: Given the use of a 3D encoder, how sensitive is the model's performance to different input molecular conformations? How effectively does it handle molecules with high conformational flexibility?
>
> *Response 11*: To demonstrate robustness to diverse molecular conformations, we evaluate Mol-LLaMA using three different conformations for each molecule, generated by RDKit and OpenBabel. The evaluation is conducted on the representative molecules in Table 3, which have an average of 11.11 rotatable bonds and thus can generate diverse conformations. As shown in Table R5, **the model’s performance remains largely consistent across conformations in totals, indicating strong robustness**. We will also add these results in the final revision.
>
> |Models|Help.|Relev.|Acc.|Details|Overall|Help.|Relev.|Acc.|Details|Overall|Help.|Relev.|Acc.|Details|Overall|
> |-|-|-|-|-|-|-|-|-|-|-|-|-|-|-|-|
> |Mol-LLaMA|1.127|1.145|1.154|1.090|1.125|1.224|1.266|1.302|1.211|1.251|1.578|1.840|2.030|1.528|1.744|
> |Diverse Conf|1.058|1.070|1.112|1.010|1.065|1.171|1.225|1.301|1.138|1.213|1.641|1.895|2.203|1.577|1.841|
>
> *Table R5: Quantitative evaluation with diverse conformations.*
>
> ---
> Due to the character limit, our responses continue below.

---

> > ### Author Response · Authors · 2025-08-07
> > **Thank you for engaging in the discussion (Part 2)**
> >
> > *Comment 12*: Additional comparison with larger models such as GPT-4.5 and other existing general-purpose (multi-modal) chemical language models.
> >
> > *Response 12*: We sincerely appreciate your valuable suggestion. Unfortunately, GPT-4.5 is not available via API and has usage limitations on ChatGPT. Therefore, we use GPT-4.1, which is the most recent accessible version apart from GPT-4.5. Additionally, we compare Mol-LLaMA with the suggested models, including ChemLLM-SFT [4], ChemLLM-DPO [4], and ChemVLM [5]. As shown in Table R6, Mol-LLaMA outperforms open-source molecular LLMs such as ChemLLM and ChemVLM. Mol-LLaMA exceeds GPT-4.1 in biological understanding, showing the effectiveness of Mol-LLaMA. We will add these results in the final revision.
> >
> > |Models|Help.|Relev.|Acc.|Details|Overall|Help.|Relev.|Acc.|Details|Overall|Help.|Relev.|Acc.|Details|Overall|
> > |-|-|-|-|-|-|-|-|-|-|-|-|-|-|-|-|
> > |ChemLLM-7B-1.5-SFT|0.226|0.239|0.176|0.178|0.211|0.263|0.285|0.220|0.204|0.248|0.321|0.348|0.317|0.289|0.320|
> > |ChemLLM-7B-1.5-DPO|0.202|0.196|0.163|0.171|0.190|0.213|0.212|0.185|0.205|0.200|0.267|0.260|0.255|0.265|0.258|
> > |ChemVLM-8B|0.214|0.240|0.165|0.129|0.205|0.220|0.236|0.172|0.133|0.208|0.241|0.263|0.290|0.188|0.248|
> > |GPT-4.1|1.153|1.122|1.184|1.181|1.159|1.188|1.184|1.229|1.238|1.205|1.320|1.396|1.564|1.342|1.407|
> > |Mol-LLaMA|1.127|1.145|1.154|1.090|1.125|1.224|1.266|1.302|1.211|1.251|1.578|1.840|2.030|1.528|1.744|
> >
> > *Table R6: Quantitative evaluation with additional baselines.*
> >
> > ---
> > [2] Levtsov, Georgii, and Dmitry Ustalov. "Confidence and Stability of Global and Pairwise Scores in NLP Evaluation." arXiv preprint arXiv:2507.01633 (2025).
> >
> > [3] Liusie, Adian, Potsawee Manakul, and Mark JF Gales. "LLM comparative assessment: Zero-shot NLG evaluation through pairwise comparisons using large language models." arXiv preprint arXiv:2307.07889 (2023).
> >
> > [4] Zhang, Di, et al. "Chemllm: A chemical large language model." arXiv preprint arXiv:2402.06852 (2024).
> >
> > [5] Li, Junxian, et al. "Chemvlm: Exploring the power of multimodal large language models in chemistry area." Proceedings of the AAAI Conference on Artificial Intelligence. Vol. 39. No. 1. 2025.
> >
> >
> > ---
> > We appreciate your time and effort in reviewing our paper and hope that our responses and additional results have addressed your concerns and provided further clarity. If you have further questions, we would be pleased to respond.

---

> > > ### Comment · Reviewer_6V81 · 2025-08-08
> > >
> > > Thank you for your detailed response. The additional experiments you provided have largely addressed my previous concerns.
> > >
> > > However, I still hold the reservation that the proposed method may not be the most effective path toward creating a truly general-purpose foundational model for chemistry. In my view, more convincing routes toward that goal would likely involve continued pre-training on large-scale chemical and biological corpora or learning from real-world feedback. For this reason, I believe the long-term significance of this specific approach is somewhat limited.
> > >
> > > Nevertheless, considering the strong performance demonstrated across multiple tasks in the manuscript, I raise my rating to 4.

---

### Official Review · Reviewer_JRtJ · 2025-06-28

**Clarity:** 3
**Significance:** 2
**Originality:** 2
**Rating:** 4
**Confidence:** 3

**Summary:**

In this work, the author proposed Mol-LLaMA which could poentially enhance the understanding of molecular features with strong reasoning and explainability. It introduces a novel instruction dataset that includes detailed structural descriptions, structure-to-feature relationship explanations, and comprehensive conversations. The model also integrates complementary information from different molecular encoders, improving the understanding of molecular structures and features. Overall, Mol-LLaMA outperforms existing models in predicting and explaining molecular properties, potentially accelerating scientific discovery in chemistry and biology.

**Questions:**

1. Use more nuanced evaluation criteria that consider both text similarity and chemical factual accuracy to provide a more balanced assessment of the model's performance.
2. Include a wider variety of molecular structures and properties in the instruction dataset to ensure comprehensive coverage like QM9 [1] or GEOM datasets.[2]

[1] Ramakrishnan R, Dral P O, Rupp M, et al. Quantum chemistry structures and properties of 134 kilo molecules[J]. Scientific data, 2014, 1(1): 1-7.

[2] Axelrod S, Gomez-Bombarelli R. GEOM, energy-annotated molecular conformations for property prediction and molecular generation[J]. Scientific Data, 2022, 9(1): 185.

**Ethical Concerns:**

["NO or VERY MINOR ethics concerns only"]

**Final Justification:**

The authors effectively addressed concerns regarding the suitability of LLMs for molecular property prediction by highlighting Mol-LLaMA’s strengths in interpretability, flexible interaction, and fine-tuning capabilities, supported by empirical results.

**Limitations:**

Yes

**Quality:**

3

**Strengths And Weaknesses:**

1. Provided chemistry dataset includes detailed structural descriptions, structure-to-feature relationship explanations.
2. The blending module integrates information from different molecular encoders, improving the accuracy of molecular analysis.
3. Mol-LLaMA outperforms existing models in predicting and explaining molecular properties, showcasing its potential to accelerate scientific discovery in chemistry and biology.

Weaknesses:
1. There is a concern about whether LLMs are fundamentally suitable for certain tasks in the benchmark, such as molecular property prediction, which is why only LLM results were presented.
2. The molecule-to-text evaluation relies exclusively on text similarity metrics (BLEU, ROUGE, METEOR), which are inadequate for assessing chemical factual accuracy.
3. The evaluation of LLMs based on helpfulness and relevance scores raises doubts about whether these scores accurately reflect the correctness of the information.
4. The multimodal model and multimodal text data for molecules appear to lack originality.

[1] Liu P, Ren Y, Tao J, et al. Git-mol: A multi-modal large language model for molecular science with graph, image, and text[J]. Computers in biology and medicine, 2024, 171: 108073.
[2] Liu S, Nie W, Wang C, et al. Multi-modal molecule structure–text model for text-based retrieval and editing[J]. Nature Machine Intelligence, 2023, 5(12): 1447-1457.

---

> ### Author Rebuttal · Authors · 2025-07-30
>
> We sincerely thank you for your constructive and helpful comments. We appreciate the positive comments.
> - Constructed instruction dataset includes detailed structural descriptions and structure-to-feature relationship explanations.
> - The proposed blending module improves the accuracy of molecular analysis.
> - Mol-LLaMA outperforms in predicting and explaining molecular properties, showing its potential to accelerate scientific discovery.
>
> We initially address your concerns below.
>
> ---
> *Comment 1*: There is a concern about whether LLMs are fundamentally suitable for certain tasks in the benchmark, such as molecular property prediction.
>
> *Response 1*: LLMs and specifically Mol-LLaMA exhibit significant advantages for scientific domains in three key areas: 1) **Interpreability**, 2) **Flexible Interaction**, and 3) **Fine-tuning Capability**.
>
> > ### Interpretability
>
> **LLMs, including Mol-LLaMA, generate interpretable outputs by detailed explanations, which is critical for scientific applications such as drug discovery where reliability and transparency are essential**. In contrast, traditional predictive models lack interpretability, limiting their practical application. Especially, Mol-LLaMA exhibits its interpretability and reasoning capabilities for molecules, providing relevant, helpful, and detailed responses as shown in Table 3 and 4. This interpretability of Mol-LLaMA can accelerate scientific discovery by fostering trust and insight.
>
> > ### Flexible Interaction
>
> **LLMs and Mol-LLaMA support interactive and adaptive use cases, such as user-driven prompting and the integration of external domain knowledge. This flexibility is valuable for addressing complex scientific problems**. One might provide additional molecule- or task-specific information via prompts to guide the model predictions, enhancing both performance and interpretability. Especially, Mol-LLaMA is beneficial in this utilization as shown in Table 4. Specifically, Mol-LLaMA shows a significant performance gain and achieves the best performance when prompted with task-specific information while maintaining helpful and interpretable responses.
>
> > ### Fine-tuning Capability
>
> **Open-source LLMs can be specialized for specific tasks via fine-tuning. In Table 13, we demonstrate that Mol-LLaMA, fine-tuned on property prediction tasks, significantly outperforms the state-of-the-art model without LLMs** (UniMol). This result highlights the promising opportunities for further utilization of Mol-LLaMA via fine-tuning. Please note that all datasets, codes, and model weights will be publicly released.
>
> ---
> *Comment 2*: The molecule-to-text evaluation relies exclusively on text similarity metrics (BLEU, ROUGE, METEOR), which are inadequate for assessing chemical factual accuracy.
>
> *Response 2*: **In the main experiments of Table 3 and 4, we do not use similarity metrics such as BLEU, ROUGE, or METEOR, but leverage the LLM-as-a-judge** to evaluate not only the factual accuracy but also the quality and informativeness by assessing the textual semantics via LLMs. **In Table 12, we strictly follow the evaluation settings of the benchmark to ensure fair comparison** with the baselines, showing the superior performance of our Mol-LLaMA in the task transfer scenarios.
>
> ---
> *Comment 3*: The evaluation of LLMs based on helpfulness and relevance scores raises doubts about whether these scores accurately reflect the correctness of the information.
>
> *Response 3*: **The helpfulness and relevance scores depend on the correctness, since these metrics inherently consider the factual correctness**. We experimentally validate these characteristics by intentionally generating factually incorrect outputs. Specifically, given a pair of the $i$-th molecule $X\_i$ and the generated structural description of the $i$-th molecule $D\_i$, we instruct GPT-4o to evaluate the quality of a permuted pair, defined as $(X\_i, D\_j)$ where $i\\neq j$. As shown in Table R1, factually incorrect outputs (Permuted) significantly degrade the overall performance, including helpfulness and relevance, indicating that these metrics are affected by the factual correctness.
>
> ||Help|Relev|Acc|Details|Overall|
> |-|-|-|-|-|-|
> |Original|1.126|1.145|1.154|1.090|1.125|
> |Permuted|0.353|0.299|0.219|0.346|0.300|
>
> *Table R1: Quantitative results on structural understanding*
>
> ---
> *Comment 4*: The multimodal model and multimodal text data for molecules appear to lack originality. [1, 2]
>
> *Response 4*: Compared to the reference works, our work made significant contributions in threefolds: 1) **dataset construction pipeline tailored to molecules**, 2) **model architecture to effectively combine the different representations**, and 3) **application-level contributions**.
>
> First, **we propose a dataset construction pipeline for molecular understanding, which is not explored in the reference works**. We observe that relying on public databases, including the datasets used in [1, 2], limits the knowledge scope of trained models, often generates irrelevant responses, makes wrong predictions, and limits interpretability [lines 46-52]. To address this limitation, from our key insight into the hierarchical relationships between molecular structures and their properties [lines 132-136], we design three data types and construct a large and comprehensive instruction dataset. Our dataset not only facilitates an understanding of general molecular features but also enhances explainability and reasoning capabilities [lines 138-159], which brings a substantial contributions compared to the multimodal text data in [1, 2].
>
> Second, **we propose a model architecture that effectively integrates the complementary information from different and essential molecular representations**. Specifically, the main contributions on the model architectures can be summarised as follows:
> - We utilize both 2D and 3D molecular representations as they have their distinct and essential information for modeling molecular structures [lines 167-172]. In contrast, the reference work [1] employs 2D molecular structures with the images of 2D molecular structures, not providing additional information from using images.
> - Our proposed blending module effectively integrates these distinct advantages between 2D and 3D molecular encoders [lines 173-178], while the reference work [1] utilizes a simple concatenation, which cannot fully leverage the complementary information from different molecular representations as experimentally validated in Table 6 (Right, 2D+3D (Concat)).
>
> Third, our work and multi-modal model, Mol-LLaMA, have distinct advantages for the practical utility as follows:
> - Model Abilities: **Mol-LLaMA can play diverse roles with users and, thus facilitate the scientific discovery, whereas the reference work [2] is specialized for several tasks** because of the limited model architecture. Please note the reference work [2] (i.e., MoleculeSTM) is one of our molecular encoders [lines 171-172].
> - Further Utilization: **Our dataset construction pipelines and model architecture can be further utilized for other biomolecules**, including RNAs, proteins, and their complexes [lines 299-309], which are fundamental and essential components to understand the living organisms.
>
> [1] Liu, Pengfei, et al. "Git-mol: A multi-modal large language model for molecular science with graph, image, and text." Computers in biology and medicine 171 (2024): 108073.
>
> [2] Liu, Shengchao, et al. "Multi-modal molecule structure–text model for text-based retrieval and editing." Nature Machine Intelligence 5.12 (2023): 1447-1457.
>
> ---
> *Comment 5*: Include a wider variety of molecular structures and properties in the instruction dataset to ensure comprehensive coverage like QM9 [3] or GEOM datasets. [4]
>
> *Response 5*: Our dataset includes 320K molecular structures derived from PubChem, providing **extensive coverage comparable to QM9 (133K) and GEOM (450K)**. However, **unlike QM9 and GEOM, which primarily annotate highly specific numerical properties** (e.g., quantum chemical properties in QM9 and BBBP or BACE-1 binding affinity in GEOM), **our dataset focuses on general molecular properties described textually**. The narrow and numeric-specific annotations in QM9 and GEOM limit their applicability for instructional purposes, as discussed in lines 118-125. By leveraging PubChem’s broad textual descriptions, our dataset effectively captures a comprehensive spectrum of molecular properties and ensures suitability for diverse instructional tasks.
>
> [3] Ramakrishnan, Raghunathan, et al. "Quantum chemistry structures and properties of 134 kilo molecules." Scientific data 1.1 (2014): 1-7.
>
> [4]  Axelrod, Simon, and Rafael Gomez-Bombarelli. "GEOM, energy-annotated molecular conformations for property prediction and molecular generation." Scientific Data 9.1 (2022): 185.
>
> ---
> We thank the reviewer for your time and feedback and hope that our responses have addressed your concerns. If you have any further questions, we are happy to address them. If your concerns are appropriately addressed, we hope you kindly consider updating the rating accordingly.

---

> > ### Comment · Area_Chair_QMDC · 2025-08-05
> >
> > Dear Reviewer JRtJ,
> >
> > Would you mind to have a check over authors' replies?
> >
> > AC

---

> > > ### Author Response · Authors · 2025-08-08
> > > **Brief Summarization of Our Rebuttal**
> > >
> > > Dear Reviewer JRtJ,
> > >
> > > We sincerely appreciate your time and effort in reviewing our work. To streamline your review, we briefly summarize our rebuttal below.
> > >
> > > ---
> > > **W1: Importance of LLMs for scientific discovery**
> > >
> > > Beyond the molecular property prediction, LLMs exhibit their capabilities to accelerate scientific discovery: 1) **Interpretability**, 2) **Flexible interaction**, and 3) **Fine-tuning capability**. Further, **our Mol-LLaMA exhibits those capabilities, also outperforming the state-of-the-art model without LLMs (UniMol) on the property prediction task.**
> > >
> > > ---
> > >
> > > **W2 & Q1: Reliance on text similarity metrics for evaluation (BLEU, ROUGE, METEOR)**
> > >
> > > **For the main experiments in Tables 3 and 4, we adopt LLM-as-a-judge to rigorously assess chemical factual accuracy by considering the textual semantics.** The text similarity metrics used in Table 12 are to strictly follow the experimental settings of the previous works.
> > >
> > > ---
> > >
> > > **W3: Whether qualitative metrics, such as helpfulness and relevance scores, adequately reflect accuracy**
> > >
> > > An additional experiment in Table R1 showed that **factually incorrect responses significantly degenerate performance**, including helpfulness and relevance scores, **indicating that these qualitative metrics reflect accuracy**.
> > >
> > > ---
> > > **W4: Regarding originality**
> > >
> > > Our work made significant contributions, including 1) **a novel dataset construction pipeline tailored to molecules**, 2) **a straightforward yet effective model architecture to combine the different representations**, and 3) **application-level contributions such as versatility and further utilization for other biomolecules such as RNAs or proteins**, which are **not covered by the suggested works**.
> > >
> > > ---
> > > **Q2: Wide variety of molecular structures and properties in the instruction dataset**
> > >
> > > **Our instruction dataset contains a large number of molecular structures** comparable to QM9 and GEOM, and **general textual properties that are well-studied in the scientific literature**, rather than the specific numerical properties in QM9 and GEOM, such as quantum properties.
> > >
> > > ---
> > > We thank you once again for taking the time to review our work and hope that this summarization helps facilitate your review process. As the end of the discussion phase is approaching, we would be grateful if you could let us know whether our responses have addressed your concerns. If there are any additional concerns or questions, we are happy to address them.
> > >
> > > ---
> > > Best regards,
> > >
> > > Authors

---

> > > > ### Author Response · Authors · 2025-08-09
> > > > **Kind Reminder and Request for Feedback on Our Rebuttal**
> > > >
> > > > Dear Reviewer JRtJ,
> > > >
> > > > As the discussion phase will end soon, we would like to kindly remind you regarding our rebuttal. We have clarified the importance of LLMs and our Mol-LLaMA, the rigorousness of the metrics used in our paper, the originality of our work, and the diversity of our instruction dataset.
> > > >
> > > > If possible, could you kindly review our rebuttal and consider providing a response to our clarifications? Your feedback would be greatly appreciated.
> > > >
> > > > Sincerely,
> > > >
> > > > Authors

---

### Official Review · Reviewer_QhZK · 2025-07-01

**Clarity:** 3
**Significance:** 3
**Originality:** 3
**Rating:** 4
**Confidence:** 4

**Summary:**

This paper introduces Mol-LlaMA, a multimodal LLM designed for general understanding and reasoning about molecules. The authors create a novel instruction dataset with three types of synthetic data: structural descriptions of functional groups, structure-to-feature relationship explanations, and conversations based on the first two, all generated using GPT-4o.

Architecture-wise, Mol-LlaMA uses 2D (MoleculeSTM) and 3D (UniMol) molecular encoders to capture structural information. These embeddings are aligned through a blending module, and a Q-Former projects the combined embeddings to the LLM.

Detailed experimental results show that Mol-LlaMA outperforms other baselines across various tasks.

**Questions:**

* Is there any evaluation performed on GPT-4o's accuracy in extracting functional groups and their connectivity information from the IUPAC notation of the molecule when generating the "Detailed Structural Description" data type?

* Please include a full ablation study on other benchmarks like property prediction, MoleculeQA, and 3D-MoIT? This could help to better understand the comprehensive benefits of the Mol-LLaMA architecture and data types.

* Is there any plan to publicly release the data and code?

**Ethical Concerns:**

["NO or VERY MINOR ethics concerns only"]

**Final Justification:**

The authors present a novel method to construct an instruction dataset for molecules, combining structural, chemical, and biological reasoning data. The experimental results are promising and highlight the potential of the method.

While some of my concerns about the evaluation with GPT-4o were addressed, the issues of dataset quality and potential hallucinations due to reliance on IUPAC naming remain unresolved.

**Limitations:**

yes

**Quality:**

3

**Strengths And Weaknesses:**

**Strengths:**

* The authors present a novel and comprehensive instruction dataset that covers structural, chemical, and biological reasoning data about the molecule.
* The proposed 2D-3D blending module effectively aligns information from different molecular encoders, and the benefits of this over single-encoder approaches are covered in the experimental results.
* Qualitative and quantitative experimental results show that Mol-LlaMA outperforms baselines on tasks like property prediction and molecular comprehension.


**Weaknesses:**

* The instruction dataset is fully synthetic and heavily relies on GPT-4o. Although GPT-4o is used to filter incorrect data, some hallucinations may still persist. A more rigorous evaluation of the dataset quality would strengthen the work.
* GPT-4o is used both for generating and evaluating the data, which may introduce bias. Including human evaluations would provide a more robust and independent assessment of Mol-LlaMA’s capabilities.

---

> ### Author Rebuttal · Authors · 2025-07-30
>
> We sincerely thank you for your constructive and helpful comments. We appreciate the positive comments.
> - The proposed instruction dataset is novel and comprehensive, covering structural, chemical, and biological reasoning data.
> - The proposed blending module effectively aligns information from different molecular encoders.
> - Experimental results show that Mol-LLaMA outperforms baselines.
>
> We initially address your concerns below.
>
> ---
> *Comment 1*: The dataset is fully synthetic, relying on GPT-4o.
>
> *Response 1*: To avoid potential misunderstanding, we would like to clarify that, while our dataset involves synthetic generation with GPT-4o, we mitigate possible hallucinations by grounding molecular feature annotations and IUPAC names using PubChem. This grounding ensures alignment with established chemical knowledge and enhances the reliability of the dataset.
>
> ---
> *Comment 2*: A more rigorous evaluation of the dataset quality beyond using GPT-4o would strengthen the work.
>
> *Response 2*: We appreciate your valuable suggestions. However, we chose GPT-4o as the primary model because, to the best of our knowledge, **it is currently the only LLM whose capabilities in molecular and scientific reasoning have been extensively analyzed and documented** [1]. At the time of our study, other LLMs lacked comparable in-depth evaluations in the molecular domain, which limited the reliability for those LLMs.
>
> To validate whether other LLMs actually help to improve the dataset quality, we further filter out data using diverse LLMs: GPT-4o, Qwne3-14B, Gemma-3-12B-IT, and Llama-3.1-8B-Instruct, and select the data with a score of 4 for all evaluator LLMs. As shown in Table R1, the quadruple filtration (Mol-LLaMA$^*$) does not bring a significant performance improvement, implying that other LLMs misjudge to remove the correct data and narrow the knowledge scope of the trained model. We will add the experimental results in the final revision.
>
> Even though the current LLMs have limitations in evaluating data quality, **our dataset construction pipeline is readily adaptable with emerging LLMs that exhibit strong domain expertise in molecular sciences, which is a notable contribution for building molecular LLMs**.
>
> |Models|Help.|Relev.|Acc.|Details|Overall|Help.|Relev.|Acc.|Details|Overall|Help.|Relev.|Acc.|Details|Overall|
> |-|-|-|-|-|-|-|-|-|-|-|-|-|-|-|-|
> |Mol-LLaMA|1.126|1.145|1.154|1.090|1.125|1.224|1.266|1.302|1.211|1.251|1.578|1.840|2.030|1.528|1.744|
> |Mol-LLaMA$^*$|1.004|1.024|1.035|0.941|1.006|1.143|1.193|1.255|1.113|1.179|1.605|1.858|2.097|1.569|1.801|
>
> *Table R1: Quantitative evaluation on structural (left 5 criteria), chemical (middle 5 criteria), and biological (right 5 criteria) understanding.*
>
> [1] Microsoft Research AI4Science, and Microsoft Azure Quantum. "The impact of large language models on scientific discovery: a preliminary study using gpt-4." arXiv preprint arXiv:2311.07361 (2023).
>
> ---
> *Comment 3*: GPT-4o is used both for data generation and performance evaluation, which may introduce bias.
>
> *Response 3*: Thank you for your valuable suggestion. To robustly evaluate the quality of generated responses, we report the average of all metrics measured by GPT-4o, Qwen3-14B, Gemma-3-12B-IT, and Llama-3.1-8B-Instruct. As shown in Table R2, **Mol-LLaMA continues to outperform all baselines, including GPT-4o, and the overall trend remains consistent with Table 3, indicating that the evaluation of other LLMs agree with the one of GPT-4o**. We will include these results in the final revision.
>
> |Models|Help.|Relev.|Acc.|Details|Overall|Help.|Relev.|Acc.|Details|Overall|Help.|Relev.|Acc.|Details|Overall|
> |-|-|-|-|-|-|-|-|-|-|-|-|-|-|-|-|
> |**Llama-2-based**|
> ||
> |Llama-2-7B-Chat|0.372|0.374|0.264|0.359|0.328|0.534|0.520|0.405|0.551|0.492|0.479|0.408|0.345|0.598|0.441|
> |Mol-Instructions|0.217|0.239|0.252|0.131|0.202|0.242|0.274|0.301|0.145|0.229|0.319|0.376|0.416|0.201|0.315|
> |LlasMol|0.278|0.306|0.304|0.204|0.265|0.295|0.335|0.312|0.215|0.280|0.332|0.378|0.443|0.260|0.341|
> |3D-MoLM|0.674|0.658|0.564|0.719|0.643|0.784|0.788|0.701|0.798|0.762|0.898|0.945|0.953|0.911|0.923|
> |LLaMo|0.306|0.407|0.443|0.179|0.312|0.345|0.464|0.524|0.197|0.356|0.454|0.630|0.802|0.236|0.490|
> |Mol-LLaMA|1.042|1.063|1.063|0.960|1.038|1.124|1.179|1.212|1.044|1.146|1.376|1.607|1.683|1.167|1.468|
> ||
> |**Llama3.1-based**|
> ||
> |Llama3.1-8B|0.669|0.680|0.560|0.648|0.634|0.698|0.693|0.600|0.695|0.667|0.705|0.663|0.617|0.732|0.674|
> |Mol-Instructions|0.261|0.334|0.369|0.152|0.263|0.265|0.350|0.396|0.149|0.269|0.358|0.468|0.577|0.191|0.367|
> |3D-MoLM|0.835|0.874|0.794|0.782|0.817|0.902|0.973|0.902|0.829|0.900|1.114|1.295|1.346|0.970|1.180|
> |LLaMo|0.480|0.616|0.560|0.301|0.473|0.401|0.534|0.560|0.245|0.414|0.596|0.788|0.834|0.311|0.603|
> |Mol-LLaMA|1.052|1.081|1.092|0.964|1.049|1.139|1.192|1.225|1.068|1.163|1.459|1.699|1.801|1.210|1.555|
> |Mol-LLaMA$^*$|1.001|1.026|1.030|0.911|0.997|1.113|1.170|1.205|1.034|1.139|1.421|1.640|1.741|1.223|1.522|
>
> *Table R2:  Quantitative evaluation on structural (left 5 criteria), chemical (middle 5 criteria), and biological (right 5 criteria) understanding using diverse evaluators.*
>
> ---
> *Comment 4*: Including human evaluations would provide a more robust and independent assessment of Mol-LlaMA’s capabilities.
>
> *Response 4*: Unfortunately, **conducting human evaluation is difficult due to the high complexity of accurately assessing molecular properties**. Accurate and reliable evaluations of model-generated responses in the scientific domain require deep interdisciplinary expertise spanning chemistry, biochemistry, biology, and pharmacology. Furthermore, models may generate correct responses that are not included in the annotated descriptions as shown in Table 11, potentially introducing evaluator bias and making an accurate assessment challenging. Consequently, gathering evaluators possessing the required interdisciplinary expertise poses significant practical difficulties.
>
> As an alternative, **we rigorously evaluate Mol-LLaMA across diverse and independent tasks, providing a robust demonstration of its capabilities**. Specifically, Mol-LLaMA demonstrates its capabilities on understanding general molecular features including structural, chemical, and biological features (Tables 3 and R2 of *Response 3*), basic and quantum property prediction (Tables 4 and 13), specific biological property prediction (Table 8), and molecular comprehension benchmarks (Tables 5 and 12).
>
> ---
> *Comment 5*: Is there any evaluation performed on GPT-4o's accuracy in extracting functional groups and their connectivity information from the IUPAC notation of the molecule when generating the "Detailed Structural Description" data type?
>
> *Response 5*: To the best of our knowledge, **there is currently no benchmark to evaluate the accuracy of understanding molecular structures with IUPAC names, as annotating IUPAC names requires specialized expertise**. Meanwhile, **we chose to use IUPAC names and GPT-4o based on the experimental analyses reported in the literature.** Microsoft Research AI4Science and Microsoft Azure Quantum teams [1] conducted extensive evaluation of GPT-4 within the scientific contexts, providing informative analyses that have not been conducted for other LLMs. Additionally, the work [1] has shown that IUPAC names are the most interpretable string representations, as their tokenization of IUPAC names provides subword semantics of molecular substructures. Therefore, our selection of both IUPAC names and GPT-4o is grounded in empirical evidence and scientific relevance.
>
> ---
> *Comment 6*: A full ablation study on other benchmarks.
>
> *Response 6*: The primary aim of the ablation study on the PAMPA task is to evaluate the knowledge and reasoning capabilities required for predicting molecular properties. Therefore, we focused on data types that significantly influence these capabilities of LLMs.
>
> We further conduct an ablation study on the task transfer scenario using MoleculeQA. As shown in Table R3, each data type contributes incrementally to performance with steady improvements observed across data types (i.e., S $\rightarrow$ S+F $\rightarrow$ Full). Table R4 further demonstrates that incorporating different molecular representations and using the blending module improves performance in total, suggesting that combining 2D and 3D representations is beneficial and that the blending module effectively integrates complementary information. Overall, the impact of data types on performance gains is greater than that of model architecture, highlighting the importance of instruction dataset quality in task transfer scenarios. We will add the experimental results in the final revision.
>
> |Method|Struct.|Source|Prop.|App.|Totals|
> |-|-|-|-|-|-|
> |S|72.74|71.41|49.36|47.27|66.84|
> |S+F|76.33|72.19|46.31|45.91|68.42|
> |Conv.|72.53|71.41|48.88|45.46|66.48|
> |Full.|77.81|75.50|49.63|49.30|70.76|
>
> *Table R3: Ablation study of data types on MoleculeQA*
>
> |Method|Struct.|Source|Prop.|App.|Totals|
> |-|-|-|-|-|-|
> |2D|72.26|73.79|49.59|47.75|67.21|
> |3D|76.20|74.42|50.94|50.45|69.93|
> |2D+3D(Concat)|76.77|75.61|49.56|48.58|70.14|
> |2D+3D(Blended)|77.81|75.50|49.63|49.30|70.76|
>
> *Table R4: Ablation study of blending module on MoleculeQA*
>
> ---
> *Comment 7*: Is there any plan to publicly release the data and code?
>
> *Response 7*: We will publicly release all materials, including data, code, and model weights, to be broadly utilized for the scientific domain.
>
> ---
> We thank the reviewer for your time and feedback and hope that our responses have addressed your concerns. If you have any further questions, we are happy to address them. If your concerns are appropriately addressed, we hope you kindly consider updating the rating accordingly.

---

> > ### Comment · Reviewer_QhZK · 2025-08-05
> >
> > Thanks for your response and ablation studies.
> >
> >
> > I understand that GPT-4's capabilities for drug discovery tasks are well studied, and using IUPAC for data generation makes sense since it's more interpretable than SMILES. However, generating structural information from IUPAC using GPT-4o is prone to hallucinations. Since this is the first step in the data generation pipeline, there's a high chance of errors and hallucinated structures in the dataset. Additional analysis - ideally by experts - on the generated data could help improve the dataset's overall quality and reliability.

---

> ### Author Response · Authors · 2025-08-07
> **Thank you for engaging in the discussion**
>
> We sincerely appreciate your insightful feedback and engagement in the discussion phase. We address your concerns below.
>
> ---
> *Comment 8*: Since this is the first step in the data generation pipeline, there's a high chance of errors and hallucinated structures in the dataset.
>
> *Response 8*: We believe that our dataset construction pipeline and the resulting dataset exhibit a significant contribution to the field as follows:
> - **Model-Agnostic Nature**: Our dataset construction pipeline is model-agnostic and can be further utilized with upcoming LLMs that exhibit advanced scientific knowledge.
> - **Further Utilization**: Our dataset construction pipeline is suited for generating instruction data of biomolecules, where domain-specific knowledge and reasoning are required.
> - **Enhancement of understanding and reasoning**: The resulting dataset improves the understanding of molecular features and enhances explainability and reasoning capabilities. As a result, our model outperforms existing molecular LLMs, including GPT-4o, highlighting the strength of our datasets.
>
> ---
> *Comment 9*: Additional analysis - ideally by experts - on the generated data could help improve the dataset's overall quality and reliability.
>
> *Response 9*: We perform an additional analysis by conducting a human evaluation for a subset of our instruction dataset. Specifically, we ask nine experts specializing in chemistry and biology to evaluate the accuracy of fifteen samples in the detailed structural descriptions of our instruction dataset. In Table R4, we report Pearson and Spearman correlation coefficients between the average expert scores and those assigned by GPT-4o. **The high correlation indicates that GPT-4o’s evaluations closely align with those of experts, showcasing the reliability of GPT-4o's evaluation. Furthermore, the samples that received a score of 4 from GPT-4o had an average expert score of 3.71, validating the quality and reliability of our instruction dataset.**
>
>
> |Pearson|$p$-value|Spearman|$p$-value|
> |-|-|-|-|
> |$0.96$|$2.68\times10^{-8}$|$0.91$|$3.43\times10^{-6}$|
>
> *Table R5: Pearson correlation coefficient between the scores of human and GPT-4o.*
>
>
> ---
> We sincerely thank you again for your time and effort in reviewing our paper, and hope our responses and the expert evaluation results address your concerns. If you have further feedback and questions, we would be pleased to respond.

---

> > ### Author Response · Authors · 2025-08-09
> >
> > Dear Reviewer QhZK,
> >
> > We would like to politely draw your attention to our recent responses regarding the reliability of our instruction dataset. In these responses, we demonstrated the reliability of our dataset through human evaluation results and clarified that both our dataset construction pipeline and the resulting dataset make significant contributions.
> >
> > As the discussion phase is coming to an end, we kindly ask that you review our latest responses. We sincerely thank you again for taking the time to review our paper.
> >
> > Sincerely,
> >
> > Authors

---

### Note · Authors · 2025-08-14

We sincerely appreciate the Area Chair and all reviewers for their time, effort, and constructive feedback in reviewing our work. We are deeply grateful that all reviewers acknowledged the following key contributions:
- (QhZK, JRtJ, 6V81, voqH) Novelty and comprehensiveness of our instruction dataset
- (QhZK, JRtJ, 6V81, voqH) Effectiveness of our model architecture in integrating information from 2D and 3D molecular encoders.
- (QhZK, JRtJ, 6V81) Superior performance of our Mol-LLaMA compared to baseline models.
- (voqH) Soundness of our experimental analysis, including ablation studies validating our design choices.

---
During the discussion, we made every effort to address all concerns raised by the reviewers, as summarized below:

1. We demonstrated the reliability of our constructed instruction dataset via a detailed explanation of GPT-4o’s advantages, complemented by experimental results including additional filtering and expert evaluation.
2. We validated the rigorousness of our experimental evaluation settings, showing that GPT-4o is not biased toward its own style or knowledge.
3. We emphasized the importance of Mol-LLaMA in facilitating scientific discovery.
4. We clarified our notable and novel contributions, including our model architecture, our dataset construction pipeline, and the resulting model, Mol-LLaMA.

---
Following the discussion phase, we are grateful that three reviewers (QhZK, 6V81, and voqH) voted for acceptance of our paper. In addition, we also reached out to Reviewer JRtJ to engage in the discussion phase. However, this reviewer did not respond.

We firmly believe that our contributions toward building a general-purpose assistant for biomolecules, along with the resulting foundational model, Mol-LLaMA, offer substantial and long-lasting value to the community. Thus, we respectfully ask that all the points outlined above be taken into consideration, and in the meantime, we will incorporate all the suggestions into the final version.

Thank you again for your time and consideration.

Sincerely,

Authors

---

### Decision · Program_Chairs · 2025-09-17

**Decision:**

Accept (poster)

**Comment:**

The paper introduces Mol-LLaMA, a large molecular language model designed to advance general molecular understanding through a novel instruction dataset and a multimodal architecture integrating 2D and 3D molecular encoders.  Reviewers generally agree on the strengths of the work, including the comprehensiveness and novelty of the dataset, the effectiveness of the proposed blending module, and the model’s strong empirical performance across multiple benchmarks. While some concerns were raised regarding reliance on GPT-4o for both data generation and evaluation, the authors provided extensive rebuttals, including additional experiments, multi-model comparisons, and limited human evaluation, which addressed most of the critical points.